# How do paediatric physical therapists teach motor skills to children with Developmental Coordination Disorder? An interview study

Ingrid P. A. van der Veer[1]*, Eugene A. A. Rameckers[1,2,3], Bert Steenbergen[4], Caroline H. G. Bastiaenen[5☯], Katrijn Klingels[1☯]

1 Rehabilitation Research Centre—REVAL, Faculty of Rehabilitation Sciences, Hasselt University, Hasselt, Belgium, 2 Department of Rehabilitation Medicine, Functioning, Participation & Rehabilitation Research Line, Research School CAPHRI, Maastricht University, Maastricht, The Netherlands, 3 Centre of Expertise, Adelante Rehabilitation Centre, Valkenburg, The Netherlands, 4 Behavioural Science Institute, Radboud University, Nijmegen, The Netherlands, 5 Department of Epidemiology, Functioning, Participation & Rehabilitation Research Line, Research School CAPHRI, Maastricht University, Maastricht, The Netherlands

☯ These authors contributed equally to this work.
* ingrid.vanderveer@uhasselt.be

**Data Availability Statement:** Following documents are open access available in the UK DataService

## Abstract

### Background

When teaching motor skills, paediatric physical therapists (PPTs) use various motor learning strategies (MLSs), adapting these to suit the individual child and the task being practised. Knowledge about the clinical decision-making process of PPTs in choosing and adapting MLSs when treating children with Developmental Coordination Disorder (DCD) is currently lacking. Therefore, this qualitative study aimed to explore PPTs' use of MLSs when teaching motor skills to children with DCD.

### Methods

Semi-structured individual and group interviews were conducted with PPTs with a wide range of experience in treating children with DCD. A conventional content analysis approach was used where all transcripts were open-coded by two reviewers independently. Categories and themes were discussed within the research group. Data were collected until saturation was reached.

### Results

Twenty-six PPTs (median age: 49 years; range: 26–66) participated in 12 individual interviews and two focus-group interviews. Six themes were identified: (1) PPTs *treated* children in a *tailor-made* way; (2) PPTs' *teaching style* was either more indirect or direct; (3) PPTs used various strategies to improve children's *motivation*; (4) PPTs had reached the *optimal level of practice* when children were challenged; (5) PPTs gave special attention to *automatization and transfer* during treatment; and (6) PPTs considered *task complexity* when choosing MLSs, which appeared determined by task constraints, environmental demands, child and therapist characteristics.

ReShare repository (10.5255/UKDA-SN-856735): 1. the codebook and memo generated during analyses, 2. the full interview guides of the individual and focus-group interviews, and 3. the informed consent forms. The interview transcripts generated and analysed during this study are not publicly available due to their containing information that could compromise research participants' privacy/consent.

**Funding:** The authors received no specific funding for this work.

**Competing interests:** The authors have declared that no competing interests exist.

## Conclusion

PPTs' clinical decision-making processes in choosing MLSs appeared strongly influenced by therapist characteristics like knowledge and experience, resulting in large variation in the use of MLSs and teaching styles to enhance motivation, automatization, and transfer. This study indicates the importance of the level of education on using MLSs to teach children motor skills, and clinical decision-making. Future research should focus on implementing this knowledge into daily practice.

## Introduction

Teaching motor skills is fundamental in therapeutic interventions for children with motor disabilities [1]. Paediatric physical therapists (PPTs) aim to improve children's motor skills by using motor learning strategies (MLSs). MLSs can be described as observable actions of a PPT enhancing motor learning [2]. In general, three categories of MLSs can be distinguished: (1) instructions; (2) feedback; and (3) organization of practice [3]. PPTs use various instructions and feedback shaped by their focus, form, frequency, timing, and information content to motivate children or to provide specific information about task performance [2]. Organization of practice concerns how they arrange exercises and materials during treatment sessions. For instance, they use random or blocked practice to increase or decrease variation between tasks [4], or variable or constant practice to increase or decrease variation within tasks [5]. PPTs can manipulate task and environment to enhance motor learning as well, for instance, by decreasing distance to the target to improve throwing beanbags into a basket [6]. See S1 File for (more) descriptions of MLSs commonly described in literature.

PPTs' use of MLS is the result of a clinical decision-making process [2]. The theoretical 'hybrid model of Developmental Coordination Disorder (DCD)' by Wilson et al. [7] advances insights into how child, task and environmental characteristics interact. It advocates that MLSs should be adapted to the unique combination of characteristics to address the specific needs and capacities of the individual child, for example, therapists simplify cognitive load by adopting less stringent task rules or using observational instructions [7]. Previous studies exploring physical and occupational therapists' use of MLSs in video-taped treatment sessions of adults or children with acquired brain injury (ABI) confirmed that they adapted their MLSs during treatment sessions [8, 9]. A think-aloud procedure with physical therapists (PTs) watching video-taped treatment sessions of themselves treating adults with ABI showed that the chosen MLS came from therapists' knowledge, observations and assessments [8]. However, in another study, investigating PTs' perspectives on the construct of motor learning, PTs stated that they had difficulty understanding the theoretical construct of motor learning, and that their knowledge was limited, steering them towards an intuitive use of MLSs [10]. Thus, PPTs need more insights into how they can choose MLSs based on the characteristics of a child, a task, and an environment.

A population with mild-to-severe problems in motor coordination and motor learning is a population with children with DCD [11]. They experience difficulties with acquiring, automatizing, and transferring motor skills (e.g. applying skills in different contexts) while having underlying deficits in predictive control, lower abilities in observational learning, and problems in motor planning [7, 12]. Their coordination and learning problems are more prominent when task complexity increases, for instance when the task has multiple steps, requires

more precision, and/or needs dual-tasking [7, 12]. As a consequence of their compromised motor abilities, these children frequently experience bullying [11], lower levels of perceived athletic competence [13], and higher levels of internalizing symptoms (e.g. depression or anxiety) [14], all resulting in lower participation levels and lower perceived health-related quality of life [11, 15].

To improve their daily motor skills, children with DCD often receive physical therapy. According to the international recommendations on the definition, diagnosis, assessment, intervention and psycho-social aspects of DCD, PPTs are advised to use evidence-based activity- or participation-oriented interventions, like Neuromotor Task Training (NTT) and Cognitive Orientation to daily Occupational Performance (CO-OP) [16]. These child-centred interventions are based on theories of motor learning, and MLSs derived from motor learning research are used to manipulate the interaction between child, task and environment to improve motor skills [17, 18]. However, a limited amount is known about the effectiveness of individual MLSs used in children with DCD. For instructions and feedback, only the effectiveness of the focus of attention (i.e. external or internal focus) has been investigated resulting in conflicting evidence [12]. For organization of practice, two studies showed no difference in the effectiveness of variable versus constant practice [19, 20].

In summary, in activity- or participation-oriented interventions the use of MLSs is considered fundamental for teaching children with DCD motor tasks. However, a limited amount is known about their effectiveness, and also about which MLSs to choose, based on characteristics of the individual child, the tasks practised, and the environment. As a first step in developing recommendations for PPTs on the use of MLSs in children with DCD, PPTs were observed and interviewed to explore their use of MLSs. These observations provided insights into PPTs' use of instructions and feedback to teach motor skills to children with DCD (5–12 years). This qualitative interview study aims to explore how the individual child and the task being practised guide PPTs' use of MLSs when teaching motor skills to these children.

## Materials and method

### Design

In this qualitative study, semi-structured individual and focus-group interviews were conducted to explore how PPTs adapt MLSs to suit children, and how the task being practised influenced their choices. The PPTs participating in the individual interviews were also observed during their therapy sessions to gain additional insights into their use of instructions and feedback. By combining interviews with observations, richer data were obtained about PPTs' use of MLSs because they could elaborate on their thoughts and choices during the interviews [21]. The results of the observations are published elsewhere [22].

The study was approved by the Medical Ethical Review Board of Maastricht University (2019–1338) for Dutch participants, and Hasselt University (CME2019/060) for Flemish participants. All PPTs gave written consent for participation after receiving written and oral information.

### Participants

Dutch and Flemish PPTs could participate if they had at least one year of experience in treating children with DCD. For the individual interviews, they were asked to videotape one of their own treatment sessions up to one week before. For the focus groups, the PPTs needed to be willing to share their experiences with colleagues.

## Procedure

**Recruitment.** PPTs were recruited between January 2020 and June 2021 in Belgium and the Netherlands, using a convenience sampling strategy [23]. A flyer was distributed within the network for PT clinical internships of Hasselt University, within two regional networks of PPTs in the southern Netherlands (Limburg and Den Bosch networks), and at several educational activities for physical and occupational therapists (e.g. symposia) in both countries. In order to collect a wide range of PPTs' perspectives, a heterogeneous sample matching the following criteria was required [23]: (1) PPTs with different backgrounds in terms of work settings (e.g. private physical therapy practice, and rehabilitation centre); and (2) variation in years of experience in treating children with DCD. The PPTs supplied their demographic characteristics (age, work setting, graduation year, and years of experience in treating children with DCD) by completing a short questionnaire.

**Individual interviews.** Individual interviews were conducted to gain insight into the individual reasons of PPTs about their choices in MLSs used to teach motor tasks to children with DCD [21, 24]. The framework described by Kallio et al. [25] was used to develop the interview guide. A preliminary semi-structured interview guide was developed by the authors who had clinical, educational, and research expertise in both motor learning and children with DCD. The interview guide started with introductory questions to get acquainted and to elicit information about PPTs' experiences in treating children with DCD. Subsequently, more specific questions explored therapists' use of MLSs with these children (Table 1). The interview guide contained suggestions for the interviewer for open-ended follow-up questions, prompts and probes which the interviewer could use to elaborate initial answers [25]. The preliminary interview guide was tested with pilot interviews to assess coverage and relevance of content, and to identify possible needs for reformulating questions and optimising the interview procedure [25]. After three pilot interviews with members of the target population, the interview guide was finalised (S2 File). The data from the pilot interviews were discarded.

The first author (IvdV) and four master's students conducted the interviews. Six students each received 35 hours of education to make them familiar with the interview guide and procedure, and to teach them in interview skills such as using prompts and probes. Education included: reading literature about interviewing, and about the topics motor learning and DCD; listening to and discussing the pilot interviews; and performing two interviews by themselves on which they received extensive feedback. Two of the six students experienced difficulties in mastering the interview skills, leaving four to conduct the interviews. The interviewers were encouraged to use the interview guide flexibly to maintain the flow of the interview [25].

**Table 1. Main topics of the individual and focus-group interview guides.**

| Individual interviews | Focus group 1 | Focus group 2 |
|---|---|---|
| PPTs' use of instructions, feedback and organisation of practice | PPTs' use of MLSs in various tasks | PPTs' adaptation of MLSs to child characteristics |
| PPTs' use of implicit and explicit motor learning approaches | The information content of instructions and feedback | PPTs' use of MLSs in various tasks |
| The adaptation of MLSs to child, task and environmental characteristics | Environmental factors guiding therapists' use of MLSs | The interaction of child, task and environment |
| | The trade-off between the child's experiences of success and failure in the intervention | |
| | The use of variation in the intervention (e.g. random practice) | |
| | PPTs' adaptation of MLSs to the child's learning stage | |

MLSs = Motor Learning Strategies

Because previous interviews showed that it was difficult for PTs to express exactly what their ideas were regarding their use of MLSs in a specific situation with a particular child [8, 10], the PPTs recorded one treatment session in which they taught motor skills to a child (aged 5–12 years) with (probable) DCD. Preferably, the child was diagnosed with DCD. However, because the mean age of receiving a diagnosis of DCD in the Netherlands is 7.02 years (SD 1.79), and the process of diagnosis takes an average of 2.79 years (SD 2.13 years) [26], PPTs were able to video-tape a treatment session of a child with probable DCD if the child met all four DSM-5 criteria for DCD. Interviewers watched the videos in preparation for the interviews and, during these, referred to situations observed to encourage therapists to elaborate on their thoughts and choices.

The audiotaped interviews lasted approximately one hour. The interviews were held in the PPTs' own workplace so that they could support their answers with demonstrations. However, due to Covid-19 restrictions, five of 12 individual interviews were conducted online with Skype or Google Meet [27].

**Focus-group interviews.**   Two focus-group interviews were planned: the first one after 10 individual interviews, and the second after the final individual interview. None of the participating PPTs had participated in the individual interviews. The focus groups were conducted in addition to the individual interviews to enhance data richness [21, 24]. The focus-group interviews: (1) deepened topics mentioned in the individual interviews; (2) clarified and elaborated on different points of view about the use of MLSs; and/or (3) determined that insights obtained from the individual interviews were shared by a larger group of therapists [21, 24]. The findings of the interview analyses prior to the focus-group interviews determined the main topics of these (Table 1). The topics of Focus group 1 emerged from the analyses of the first 10 individual interviews. The topics of Focus group 2 were modified after analysing Focus group 1 and more individual interviews. The structure of the focus-group interview guides was similar to that of the individual one: (1) an introductory question to get acquainted with each other; and (2) specific questions addressing the main topics. The focus-group interview guides (S3 File) were discussed and fine-tuned within the research team (IvdV, KK, ER, CB) to ensure the relevance and completeness of their content [21].

To moderate group discussions, the interviewer (IvdV) asked follow-up questions, used prompts and probes, and invited participants to share their thoughts. Furthermore, an assistant made notes, managed time, and ensured that all topics were discussed [21]. Each focus group had 6 to 10 participants, and was organized in a venue proposed by the participants [21]. The audiotaped focus-group interviews lasted approximately two hours.

## Data collection

Data collection started with individual interviews. An iterative process of data collection and analysis was used to sharpen the focus of the interviews as data collection progressed [28]. As a consequence, the interviews conducted after Focus group 1 focused more on how therapists adapted their MLSs to characteristics of child, task and environment. Focus group 2 was organized when data saturation in the individual interviews seemed reached. This was the case when two consecutive individual interviews identified no new themes, and provided no new meaningful information to better understand the identified themes [23, 29]. A previous interview study exploring how PTs perceive motor learning in their practice reached saturation after 12 individual interviews [10] and therefore it was expected that 12–15 individual and two focus-group interviews would be sufficient.

All research documents were coded, after removing identifying information to guarantee participants' privacy. Only one researcher (IvdV) had access to the code-key. The Flemish data and copies of the Dutch data were stored on the secured server of Hasselt University.

## Data analyses

Median age (with range), gender and nationality, and range of years of experience in treating children with DCD were reported.

The qualitative analyses used a conventional content analysis approach [30], using ATLAS.ti Windows (version 22.0.6.0) [31]. Each individual and focus-group interview was transcribed verbatim without the identifying information. The analyses followed six steps: (1) listening to the audio-tape, and reading the complete transcript, to obtain a sense of the whole; (2) line-by-line reading of the transcript, while making notes about first impressions and thoughts; (3) highlighting relevant text fragments; (4) coding these fragments using an inductive coding strategy (i.e. coding without predefined codes); (5) sorting open codes into categories; and (6) identifying themes by organizing and grouping categories into meaningful clusters [30, 32].

Steps 1 to 4 were conducted by two reviewers independently. The first six individual interviews, and both focus-group interviews, were analysed by the first author (IvdV), together with a master's student. Three students each received 25 hours of education to acquire analysing skills and to standardize the procedure. Education included: reading literature and watching YouTube videos about analysing qualitative data; and analysing two transcripts according to the described procedure, on which they received extensive feedback. The remaining six individual interviews were analysed by two students, and checked by the first author. Differences were discussed between both reviewers until consensus was reached. In case of disagreement, differences were discussed with the other reviewers. For Steps 5 and 6, multiple meetings were organized with the research group (IvdV, KK, ER, CB), comprising researchers with clinical and methodological expertise.

## Results

### Process of recruitment and data collection

After receiving the flyer, 41 PPTs and two groups of PPTs (that meet four times a year for peer-review to optimise functioning in clinical practice) requested more information about the study. Of the PPTs interested in participating in the individual interviews, six had no opportunity to videotape a treatment session, which resulted in 12 PPTs and two groups participating in the interviews (Fig 1).

After 10 individual interviews, Focus group 1 was organized with eight PPTs to elaborate on six topics gained from findings of the individual interviews (Table 1). Another two individual interviews resulted in no new themes, and provided no new meaningful information for a better understanding of the identified themes. Focus group 2 with six PPTs confirmed saturation (Fig 1).

### Participants

The PPTs had a median age of 49 years (range 26–66), with experience in treating children with DCD ranging from 4 to 40 years. Twenty-three of them worked in a private physical

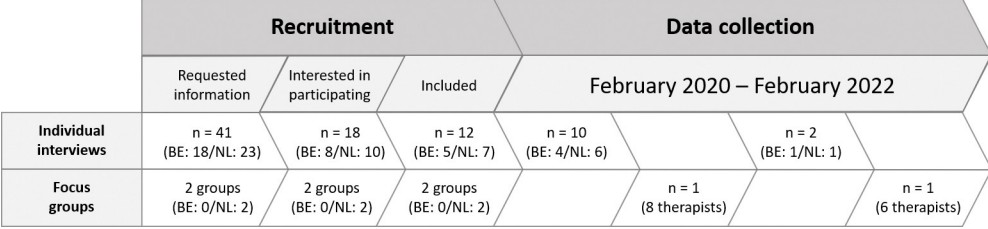

**Fig 1. Flow of the recruitment and data collection.** n = number; BE = Belgium; NL = the Netherlands.

**Table 2. Characteristics of participants.**

| | Individual interviews n = 12 | Focus group 1 n = 8 | Focus group 2 n = 6 |
|---|---|---|---|
| Age: median in years (range) | 49.5 (26–63) | 51.5 (26–61) | 47.5 (42–66) |
| Sex: f/m | 12/0 | 7/1 | 6/0 |
| Experiences in treating children with (probable) DCD: range in years | 4–40 | 4–39 | 7–20 |
| Work setting: P/RC/C | 7/3/2 | 5/0/3 | 5/0/1 |

n = number; f = female; m = male; BE = Belgium; NL = the Netherlands; P = private physical therapy practice; RC = rehabilitation centre; C = combination of private physical therapy practice and rehabilitation centre.

therapy practice, of which six combined this with working in a rehabilitation centre as well. The remaining three PPTs worked in rehabilitation centres (Table 2).

## Findings of the interview analyses

Most PPTs experienced challenges expressing their thoughts and choices during the interview. They supported their answers with many examples on using instructions, feedback and organization of practice in specific cases. Concepts, such as implicit and explicit motor learning, and specific learning strategies, like errorless learning and analogy learning (S1 File), were not explicitly known by the majority of the PPTs. However, examples showed that in fact they all used these in their daily practice. The only exception was motor imagery, which was not used at all. After the interview, most PPTs mentioned that they found the reflection on their actions valuable in optimising their use of MLSs. Six themes emerged from the analysis, providing insights into how PPTs adapted MLSs to suit child and task during treatment sessions of children with DCD. These were: (1) Tailor-made treatment; (2) Therapists' teaching style; (3) Motivation; (4) Optimal level of practice; (5) Automatization and transfer; and (6) Task complexity (Table 3). The following paragraphs will elaborate on these separate themes.

## Theme 1: Tailor-made treatment

This theme consisted of five categories: (1) interaction of child, task and environment; (2) intuition; (3) the search for "what works"; (4) PPT characteristics; and (5) child and task characteristics guiding PPTs' choices (Table 3).

All PPTs provided tailor-made treatments to children with DCD. They pointed out that the **interaction of child, task and environment** most guided their use of MLSs:

> "*If I look at a child and I see that it is anxious, then that determines how I build my track with exercises. However, if I have a parent that is scared that the child will fall, and reacts negatively every time I let the child jump* [of a height], *then that will also influence my choices. Furthermore, if a child gets demotivated due to failure, I will change the task. So, I think there is not one* [characteristic that is most relevant in making choices].*"

But they also acknowledged choosing MLSs mainly through **intuition**:

> "*It is when you ask all those questions that I start thinking about it. Because normally you just do things.*"

**Table 3. Themes, categories and quotes.**

| Themes | Categories | Quotes |
|---|---|---|
| **Tailor-made treatment** | Interaction of child, task and environment | "*I think, that in the early stage of the treatment period, child characteristics guide my choices most. Throughout the treatment period, environmental characteristics get more important.*" |
| | Intuition | "*I think, that I use that* [questions to provide feedback] *not that often, but that is not a conscious choice*" |
| | The search for "what works" | "*Some children prefer stories, while others learn more from pictures. So, you try and experience what works best.*" |
| | PPT characteristics | "*I think, that I do that a lot automatically* [using visual cues], *especially, because we also work a lot with children with autism.*" "*So, it is second nature.*" |
| | Child and task characteristics guiding PPTs' choice | "*In children with also autism I use less verbal language, but mostly demonstrations*" |
| **Therapists' teaching style** | Indirect/direct style | "*I want children to find their own movement solution.*" "*In my opinion, it* [the solution] *sticks longer.*" |
| | PPT characteristics | "*Now, I think it is mostly because it suits me* [explicit instructions]." |
| | Child characteristics | "*I want them to think about other movement strategies. Specifically, because children with DCD have rigid strategies, and you want them to try other strategies than the one that is not successful.*" |
| **Motivation** | Motivation as a prerequisite for learning | "*If they* [children with DCD] *fail too often, then I will lose them* [they will not practice anymore]." |
| | Child characteristics | "*They* [children with DCD] *are very often insecure about themselves, it is not necessarily fear, mostly they have low self-esteem. Or at least they think: I cannot do that.*" |
| | Strategies to improve motivation | "*I will give them* [children with DCD] *confidence by saying 'if I say that you can do it, you can do it'.*" "*So, I give them faith, but you have to make sure that the level of difficulty of the exercises provides them with experiences of success.*" |
| **Optimal level of practice** | Experiences of success and failure | "*I do that* [leaving room for mistakes], *to learn from mistakes.*" |
| | Learning stages | "*If the child is in the associative learning stage, and I want to reach the autonomous stage, I will use dual tasks*" |
| **Automatization and transfer** | Specific learning disabilities of children with DCD | "*Every context is unique to them* [children with DCD], *that is difficult for them.*" |
| | Strategies to enhance automatization | "*Of course it is very important that everybody does it in the same way, otherwise the child gets confused.*" |
| | Strategies to enhance transfer | With respect to using a child's own materials: "*At the end, I want them* [children with DCD] *to ty their own shoe, and not only a shoe in front of them on a table.*" |
| **Task complexity** | Task constraints | "*Tying shoe laces has a fixed sequence of steps, while ball skills are more context dependent.*" |
| | Environmental demands | "*Writing in a busy class room or in a one-to-one situation.*" |
| | Children experience tasks as difficult to learn | "*Well, the experienced level of complexity of the task* [riding a bike] *will be different for the one than the other: a child with balance problems will experience riding a bike as more complex.*" |
| | PPTs experience tasks as difficult to teach | "*In practicing ball skills it is easy to vary: catcher and thrower stand still, one of them can move, and both can move. But when practicing writing, I found it more difficult to vary.*" |

DCD = developmental coordination disorder

Several PPTs described how, in some cases, it was **a search to discover which MLSs worked best**. Their main reason for trying different MLSs was that children were experiencing difficulty mastering tasks with one MLS:

"*I experienced with this child* [the child of the video-taped treatment session], *that he did not showed improvement. Therefore, I decided to tell him exactly what I expected of him* [in the motor performance].*"

After elaborating on the clinical decision-making process in the interviews, it appeared that not only did the interaction of child, task and environment guide PPTs' choices, but that this process was influenced by **PPT characteristics** as well. Characteristics such as knowledge, preferences, experiences, character, and/or beliefs affected their choice of MLSs. For instance,

one PPT stressed the importance of pedagogical aspects within the learning environment, describing how she addressed this during treatment:

> "*I find it really important what the pedagogical context is for a child. So, does a child feel safe within the treatment and does it have autonomy? I find this important because it supports the child's development.*"

Several **child characteristics** were mentioned when PPTs elaborated on their choice for MLSs. However, variation in preferred MLSs for specific child characteristics was large. Some of the child characteristics will be discussed in more detail in the next themes. Following child characteristics were mentioned frequently: (1) deficits in executive functions (Theme 2); (2) level of motivation (Theme 3); (3) level of perceived competence (Theme 3); (4) presence of movement anxiety (Theme 3); (5) learning stage (Theme 4); (6) presence of comorbidity (e.g. autism spectrum disorder); and (7) age. Cognition, behavioural aspects like resistance, and verbal capacities were mentioned by some therapists. For **task characteristics**, their complexity seemed an important guiding characteristic (Theme 6).

**Theme 2: Therapists' teaching style.**   This theme consisted of three categories: (1) indirect/direct style; (2) PPT characteristics; and (3) child characteristics (Table 3).

In general, two types of teaching styles could be recognized: **indirect and direct styles**. A greater part of the PPTs preferred an indirect style, using questions and/or manipulations of task and environment to guide children with DCD to the correct movement solution. However, some of them preferred a direct style, instructing children exactly what to do. For instance, one PPT talked about how she used demonstrations with extensive verbal guidance to improve jumping skills:

> "*For example, I demonstrate jumping while also guiding very verbally.*" "*I give that guidance, so that he takes over from me, and starts guiding himself, first out loud and eventually in his head.*"

Reasons for preferring an indirect or direct style differed. Some were related to **PPT characteristics**: (1) it suited them because it is the style they prefer themselves or it matches their character; and (2) they had learned by experience or education that a certain style works best. For example, several PPTs argued that they preferred asking questions because they believed that children learn more when discovering their own movement solution:

> "*I prefer asking questions rather than giving feedback, because I think that it* [the movement solution] *sticks better when the child comes up with it itself.*"

Other reasons were related to **child characteristics**. Some PPTs suggested that children needed to be of a certain age and stage of cognition to process the explicit instructions used in a direct style. Furthermore, all PPTs agreed that children with DCD experience problems in executive functions (e.g. motor planning, finding movement solutions, and reflecting on their own actions). For some of them, that is why they use an indirect teaching style, so that such children learn these cognitive skills by themselves. However, for others, this justified a direct teaching style, because they believed these children to be insufficiently capable of learning these cognitive skills:

> "*I do not use that* [guiding with questions] *in children with DCD, I do it with my other children, but not with them because they cannot reflect on their motor disabilities.*"

**Theme 3: Motivation.**   This theme consisted of three categories: (1) motivation as a prerequisite for learning; (2) child characteristics; and (3) strategies to improve motivation (Table 3).

All PPTs explained how a child's motivation guided their use of MLSs. A general assumption was that demotivated children will learn less. Several PPTs stressed the importance of **motivation as a prerequisite for learning**:

"*Because success makes happy and positive, and, with positive experiences, learning improves, right? That is* [scientifically] *demonstrated.*"

Furthermore, PPTs talked about how various **child characteristics** impacted a child's motivation according to their opinion. They underpinned the problems in automatization and transfer (Theme 4), and the lower levels of perceived competence of children with DCD as main reasons for these children not being motivated to practice, and getting frustrated when experiencing tasks as being too difficult:

"*If it is really difficult, and it goes wrong every time, I don't think they* [children with DCD] *will practice.*"

Other reasons mentioned were movement anxiety and behavioural aspects like resistance when children were experiencing a bad day or were fatigued.

The PPTs suggested various strategies to improve children's motivation (Table 4). For instance, one PPT talked about using themes to improve motivation:

"*The boy had no motivation, because he was playing when he had to come to me. I asked what he was doing, and he told me he was making a marble run. So, we drew marble runs when practicing writing readiness skills.*"

**Theme 4: Optimal level of practice.**   This theme consisted of two categories: (1) experiences of success and failure; and (2) learning stages (Table 3).

The majority of the PPTs emphasized that the optimal practice level is when children are challenged, but most trials are still successful. They argued that if tasks are too easy or too difficult, children will not learn and will become demotivated. They talked about the relevance of **experiences of success and failure**. Most PPTs underlined the importance of success in children with DCD:

**Table 4. Strategies to enhance children's motivation.**

| Strategies to enhance motivation |
|---|
| Giving frequent compliments and/or small rewards (e.g. stickers) enhancing self-confidence |
| Alternating "learning tasks" with "fun tasks" rewarding good practice to enhance self-confidence |
| Involving other children (e.g. friends) during treatment to increase enjoyment, and enhancing self-confidence |
| Decreasing the level of difficulty of the exercise to increase experiences of success |
| Increasing the level of difficulty between exercises more gradually, increasing experiences of success |
| Using fewer verbal instructions and feedback, and more visual cues or manipulations of task and environment, decreasing the focus on errors |
| Providing choice (e.g. in materials or exercises) to enhance autonomy |
| Working with themes that suit children's interests, increasing enjoyment |
| Changing teaching approaches (e.g. using more fantasy or competition), increasing enjoyment |

"*You make sure that the child still can perform the exercises, and that the challenge is there. But I think that, in these children, it is even more important that they get positive experiences.*"

However, several PPTs also talked about how errors enhanced learning:

"*Sometimes, you have to do something wrong to know how you should actually do it.*"

One of the child characteristics frequently considered when estimating the optimal practice level was the child's **learning stage** (e.g. cognitive stage). However, perspectives on the use of MLSs within learning stages differed. In the early stage of learning, some therapists said they used more explicit instructions and feedback, while others strongly preferred using manipulation of tasks and environment without instructions and feedback. In one focus group, PPTs discussed the use of variation in the early stage of learning. Some of them reduced variation during practice to accelerate learning, while others deliberately introduced variation because of the varying contexts found in daily life. In reaction to a PPT that elaborated on how she used various types of ball to stimulate a child's anticipation abilities in throwing, another PPTs said:

"*I practice the basics of the skill* [throwing] *to make a child familiar with it, and start varying in a later stage.*"

As learning progressed, PPTs agreed more on increasing variation and using dual tasks to enhance transfer (Theme 4). However, some PPTs said they still used explicit instructions and feedback as well to optimize performance, while others did not use these in the later stages.

**Theme 5: Automatization and transfer.** This theme consisted of three categories: (1) specific learning disabilities of children with DCD; (2) strategies to enhance automatization; and (3) strategies to enhance transfer (Table 3).

Most PPTs referred to the problems with automatization and transfer of skills as the **specific learning disabilities of children with DCD**. One PPT said:

"*Automaticity takes much more time, so it is really important to give therapy in the best way in order to automatize* [skills] *as optimally as possible.*"

The same PPT stressed that children with DCD have to keep practising tasks, as otherwise they forget how to perform them.

The PPTs suggested various **strategies to enhance automatization**. They stressed the importance of instructing parents and teachers to practise in daily life, and underpinned using the same wording in instructions and feedback:

"*They* [parents and teachers] *should use the same wording as I do, because otherwise they* [children with DCD] *will never automatize.*"

Furthermore, they suggested to practice tasks in similar ways throughout the various treatment sessions, and to decrease instructions and feedback when learning progresses to increase time for repetitions. They felt that with motivated children it was easier to achieve greater time on task.

PPTs also talked about their **strategies to enhance transfer**. For instance, they varied spatial and temporal constraints during practice (e.g. by continuously changing throwing

direction and/or speed to improve the child's catching abilities) to enhance anticipating to variable contexts in daily life:

> "*When they* [children with DCD] *know the movement pattern, than you should start changing to try to simulate other situations* [from daily life]."

Other suggested strategies were: (1) simulating daily context by using dual tasks, or inviting other children to participate during treatment; (2) practising tasks that fit the child's needs; and (3) using regular tools from children's daily life (e.g. the child's own bike).

**Theme 6: Task complexity.** This theme consisted of four categories: (1) task constraints; (2) environmental demands; (3) children experience tasks as difficult to learn; and (4) PPTs experience tasks as difficult to teach (Table 3).

Most PPTs compared two types of tasks while elaborating on how tasks guide their use of MLSs. Frequently used examples were writing, cycling, rope skipping, and tying shoe laces. These tasks were compared to throwing, catching, running, and climbing. Because PPTs found it difficult to explain exactly how these tasks differed, this topic was extensively discussed in both focus groups. The overarching theme seemed to be task complexity, with four variables determining this identified: (1) task constraints; (2) environmental demands; (3) child and (4) therapist characteristics.

The PPTs mentioned following **task constraints** making tasks more complex: (1) multiple sequential steps; (2) dual tasking; (3) specific timing requirements; (4) bimanual coordination with both hands having different functions; and (5) the requirement to follow rules, for instance, in games. For instance, one PPT said:

"*Eating is a bimanual skill, the hands must support each other, while doing opposite tasks*" "*I think that is what makes eating complex.*"

They also pointed out that **environmental demands** could increase complexity, for example riding a bike in traffic is much more complex then riding a bike on an empty schoolyard:

> "*The child could ride a bike inside very well, but he refused to ride outside.*" "*Riding a bike depends on the person or the environment.*"

Some PPTs noted that children may experience specific tasks as more difficult to learn:

> "*I find it* [rope skipping] *not difficult to teach, but I find it difficult to learn for the child* [with DCD]."

Finally, some PPTs experienced tasks as more difficult to teach, which seemed to be related to their knowledge and experience:

> "*I find skipping* [as locomotion] *very difficult to teach to a child, probably one of the most difficult tasks.*"

PPTs' opinions on how tasks guided their use of MLSs varied. In both focus groups, they discussed how they used MLSs to improve performances of complex tasks (e.g. cycling or rope skipping). MLSs varied from explicit instructions, in which the child was told step by step how to ride a bicycle, to implicit strategies with manipulations of task and environment without using instructions and feedback, for example by letting the child ride the bicycle of a hill to increase speed.

## Discussion

This qualitative study explored how PPTs adapted MLSs, based on characteristics of a child with DCD, and the task practised. One of the main findings was that PPTs intuitively choose

MLSs, and that their clinical decision-making process was not only guided by child and task, but also by their own characteristics (Themes 1 and 6). Another finding was that PPTs used indirect or direct teaching styles, and that they had different justifications for choosing a specific style in children with DCD (Theme 2). Lastly, some general key elements for motor learning in children with DCD emerged when PPTs elaborated on how child characteristics influenced their choices: (1) motivation (Theme 3); (2) optimal level of practice (Theme 4); (3) sufficient time spent on task (Theme 5); and (4) stimulating transfer (Theme 5).

## Factors guiding PPTs' process of clinical decision making

**PPT characteristics.**   Most PPTs experienced difficulties putting their thoughts into words about which characteristics led their clinical decision making. They stressed that the interaction of child, task, and environment guided their choices, as suggested by the 'hybrid model of DCD' [7]. However, the results of this study showed that PPTs choose MLSs intuitively, and that their clinical decision-making process was influenced by their own characteristics like knowledge, preferences, and beliefs as well. These findings are in line with a previous interview study that explored PTs' perspectives on the construct of motor learning and their experiences of its implementation in clinical practice [10]. PTs stated that their use of MLSs was guided by intuition, and that their limited knowledge was an important barrier to implementation [10]. The importance of knowledge has also been demonstrated in several think-aloud studies investigating PTs' clinical decision-making processes in rehabilitation, showing that knowledge from prior clinical experience, education, scientific research, and mentors or colleagues influenced their clinical decision making [8, 33, 34]. For optimal clinical decision making, PPTs require knowledge about: the use of MLSs to teach motor tasks (including adapting MLSs to child and task); the learning disabilities and associated problems of children with DCD; and basic knowledge about child development. The results of this study indicate the importance of the level of education on these topics. Specifically, PPTs' knowledge about implicit and explicit motor learning approaches appeared limited. A need for more education has also been stressed by previous research [8, 10].

**Child characteristics.**   PPTs elaborated on how specific characteristics of a child guided their use of MLSs. Various child characteristics were identified. However, because of large variation in suggested characteristics and preferred MLSs, more research is required to gain insights into how the identified characteristics can guide PPTs' decisions. One child characteristic that PPTs frequently mentioned guiding their choice of an indirect or direct teaching style was the presence of deficits in executive functions. As a result of deficits in inhibitory control, working memory, and attention, children with DCD have problems in planning and organizing activities of daily life [7, 12]. Based on their assumption whether executive functions could be trained or not, some PPTs preferred an indirect teaching style, while others preferred a direct style. Research shows that executive functions in children can be improved by training [35, 36]. Advancing critical and creative thinking, and problem-solving in movement situations encourages the development of executive functions [37]. Using questions that require children to think about movement solutions and then debriefing them about their actions is a frequently used strategy. Another is to place children in movement situations that challenge them to think about movement solutions [37]. Both strategies were used by the PPTs with indirect teaching styles. Because executive functions are important in many daily life activities (e.g. in learning at school, and in social interactions), and can be trained, PPTs can adopt an indirect teaching style to enhance the development of these executive functions when teaching motor tasks [37, 38]. PPTs choices based on the characteristics learning stage, the presence of learning disabilities, level of motivation, and level of perceived competence will be discussed in the section 'Key elements in motor learning'.

**Task complexity.** Task complexity was identified as the most important task characteristic guiding PPTs' choice for MLSs. It appeared a complex construct described by four variables: (1) task constraints; (2) environmental demands; (3) child characteristics; and (4) therapist characteristics. The 'challenge point framework' conceptualizes complexity as a result of the combination of child, task and environment to which the PPTs should adapt their MLSs, which is in line with the results of our study [39]. The framework distinguishes two types of difficulties: the nominal task difficulty is defined by the task constraints, and is considered to reflect a constant amount of difficulty; the functional task difficulty is determined by the experiences of the individual (e.g. novices experience tasks as more difficult than individuals who have already performed those tasks) and environment (e.g. throwing outside in windy circumstances is more challenging than throwing indoors) [39]. This study also demonstrated that PPTs can experience teaching specific motor tasks as more or less difficult based on their knowledge and experiences.

PPTs' opinions on which MLSs to use in complex tasks (e.g. riding a bike, tying shoe laces, and writing) differed: some used specific instructions focusing on the planning of these motor tasks, while others chose to provoke the correct movements by manipulating task and context. According to the international DCD recommendations, evidence-based methods like CO-OP and NTT can be used to teach motor tasks to children with DCD: CO-OP focuses mostly on motor planning, while NTT focuses on manipulating task and context [16–18]. Some therapists explained that they chose specific MLSs because they were trained in CO-OP or NTT. However, other reasons for choosing to focus on motor planning in complex tasks were given as well, demonstrating that PPTs own characteristics influenced their choices: (1) PPTs did not know how to manipulate complex tasks and its context; (2) it suited their own preference in learning; and (3) they believed that children needed to learn motor planning to advance learning in daily life.

## Key elements for motor learning

In the treatment of children with DCD, PPTs gave specific attention for children's motivation (Key element 1), the optimal practice level (Key element 2), adequacy of time on tasks (Key element 3), and transfer (Key element 4). The key elements motivation and the optimal practice level were related: if the practice level was too difficult or too easy, motivation decreased and learning was hampered. PPTs considered the child's learning stage when estimating the optimal practice level but their opinions on the use of explicit instructions and feedback in the early learning stage differed: some argued that children needed explicit information to learn tasks that they had not yet mastered, while others said that they reduced the amount of explicit information given because children with DCD experience difficulties with processing large amounts of information. Studies investigating effectiveness of explicit and implicit instructions and feedback used to teach functional motor skills to inexperienced children with DCD found conflicting evidence [40–42]. Systematic reviews investigating the effectiveness of these types of instructions and feedback in children with and without motor disabilities also found conflicting results [43, 44]. Thus, both explicit and implicit instructions might be used.

PPTs stressed that attention to motivation is specifically needed in children with DCD, because most children experience problems in learning motor tasks and have lower levels of perceived competence. Research confirms that both characteristics are prominent in children with DCD [12, 13]. The role of motivation in enhancing motor learning is conceptualized in the 'Optimizing Performance through Intrinsic Motivation and Attention for Learning' (OPTIMAL) theory [45]. According to this, motivation will be improved by giving autonomy to children, and by enhancing their self-confidence [45]. The findings of the current study

showed a large variation between PPTs in strategies used for improving motivation (Table 3). All of them used positive encouragements and experiences of success to enhance self-confidence, in line with the OPTIMAL theory. Only a few enhanced autonomy by giving children choice. Furthermore, some stressed the importance of enjoyment to increase motivation. A systematic review investigating effectiveness of MLSs related to the OPTIMAL theory that enhanced children's motivation showed that, in most included studies, motor performance improved more when MLSs that enhanced motivation were used compared to MLSs that did not [44]. However, no such studies were performed in children with DCD. Furthermore, the authors reported that: (1) most investigated MLSs focused on feedback; (2) not all MLSs investigated had good ecological validity; and (3) effectiveness seemed modified by child characteristics like motor abilities, and the task practised. They recommended that future studies should explore how MLSs enhancing motivation could be integrated into children's motor learning [44]. The suggested MLSs in this study could be informative for researchers investigating effectiveness of MLSs.

PPTs considered the key elements adequacy of time on tasks and transfer important during treatment, because of the specific learning disabilities of children with DCD, and their consequences on the level of participation. Again, the MLSs suggested to improve time spent on task and transfer varied widely between PPTs. Most PPTs highlighted the importance of instructing parents and teachers to practise in the child's daily context, which is in line with the international DCD recommendations [16]. These recommendations also stress to practice meaningful activities fitting children's needs, and to consider practising in small groups [16]. Both were mentioned by PPTs as strategies to enhance transfer. Furthermore, PPTs frequently mentioned using variation in practice to enhance transfer, specifically as learning progressed. Studies in children with DCD showed no differences in the effectiveness of variable versus constant practice on immediate transfer tests after Wii Fit training [19, 20]. However, a systematic review including a meta-analysis concluded that effectiveness of variable practice in predominantly healthy young adults seemed promising, but that the included studies were at a high risk of bias, had small sample sizes and were difficult to compare due to large amounts of heterogeneity [5]. The authors also mentioned that variable practice can increase enjoyment, and that it suits real-world contexts better [5]. Both arguments were also raised by therapists in our study.

## Strengths and limitations

This study is the first to explore PPTs' thoughts on how to adapt MLSs during treatment in children. The design had several strengths. Firstly, individual interviews were combined with focus-group interviews, which advanced the understanding of themes like task complexity. Furthermore, it showed that the pre-identified themes were shared by a larger group of PPTs [24]. Secondly, despite the rather small sample size, a heterogeneous group of PPTs was included with wide range of experience in treating children with DCD in different settings (Table 2) which enlarged the various perspectives. Thirdly, video-taped treatment sessions facilitated the interviews because PPTs could elaborate their thoughts more easily by referring to their own treatment session. Fourthly, all transcripts were coded by two reviewers, and the themes discussed within the research team (which comprised researchers with clinical and methodological expertise). There were also some limitations. Recruitment was challenging for several reasons: (1) PPTs found participation too time-consuming; (2) for a period of time, they had no opportunity to videotape treatment sessions because they were not allowed to treat children due to Covid-19; and (3) Covid-19 safety regulations necessitated remote participation by PPTs in individual and focus-group interviews. Further, despite intensive

recruitment efforts, a Flemish focus group could not be included. Also, only one male PPT was included in this study, which seemed a logical consequence of less males working as PPTs. In the Netherlands, only 6% of the PPTs registered in the quality system of the Dutch association of PTs was male [46]. Our research aim was to explore therapists' use of MLSs, we did not focus on differences between subgroups of PPTs. Despite the recruitment challenges, we had been able to include a heterogeneous sample of PPTs, and reached saturation. Lastly, all PPTs treated children with other diagnoses (e.g. cerebral palsy or intellectual disabilities) with whom they also used MLSs. Therefore, there is a small chance that some of the experiences shared were influenced by experiences with other types of children.

## Recommendations for future research

This study indicates the importance of the level of education on: using MLSs to teach children motor tasks (including adapting MLSs to child and task), the learning disabilities and associated problems in children with DCD, and child development. Future research should focus on implementing this knowledge into daily practice, for instance, by developing an online module about the use of MLSs with a focus on clinical decision making. Previous research has shown that an evidence-based online DCD module tailored to PTs' needs, with information about identifying, assessing and treating children with DCD, appeared relevant, applicable and useful [47]. Furthermore, it enhanced PTs' self-reported knowledge and skills, and supported evidence-based practice [48]. In order to implement knowledge effectively, systematic approaches like the knowledge translation framework should be used [49].

## Conclusions

In conclusion, this study has advanced insight into PPTs' use of MLSs in children with DCD. PPTs assumed that only the interaction of child, task, and environment guided their clinical decision making, but in reality it appeared that this process was strongly influenced by their own characteristics, namely their knowledge, experiences and beliefs. These characteristics also influenced the clinical decision-making process of choosing specific teaching styles. Because of deficits in executive functions, an indirect teaching style might have been more effective but this was not always chosen [36, 37]. Furthermore, the variation in MLSs used to enhance the child's motivation, automatization and transfer appeared large, with some choices the result of limited knowledge. The findings of this study might be of interest for treatment decisions in other populations with and without motor disabilities because the identified child characteristics are generic and the process of clinical decision making is comparable.

## Supporting information

**S1 File. Motor learning strategies commonly described in literature.**
(DOCX)

**S2 File. Interview guide for the individual interviews.**
(DOCX)

**S3 File. Interview guide for the focus-group interviews.**
(DOCX)

## Acknowledgments

We are grateful to the PPTs who participated in this study. We also would like to thank the master's students of the Master's in Rehabilitation Sciences and Physiotherapy of Hasselt

University (Belgium), and of the Master's in Paediatric Physical Therapy of Avans+ (the Netherlands) for participating in conducting the interviews and analysing the data. Furthermore, we acknowledge Les Hearn (les_hearn@yahoo.co.uk) for proofreading and editing this article.

## Author Contributions

**Conceptualization:** Ingrid P. A. van der Veer, Eugene A. A. Rameckers, Caroline H. G. Bastiaenen, Katrijn Klingels.

**Formal analysis:** Ingrid P. A. van der Veer, Eugene A. A. Rameckers, Caroline H. G. Bastiaenen, Katrijn Klingels.

**Investigation:** Ingrid P. A. van der Veer.

**Methodology:** Ingrid P. A. van der Veer, Eugene A. A. Rameckers, Caroline H. G. Bastiaenen, Katrijn Klingels.

**Project administration:** Ingrid P. A. van der Veer.

**Supervision:** Eugene A. A. Rameckers, Caroline H. G. Bastiaenen, Katrijn Klingels.

**Writing – original draft:** Ingrid P. A. van der Veer.

**Writing – review & editing:** Ingrid P. A. van der Veer, Eugene A. A. Rameckers, Bert Steenbergen, Caroline H. G. Bastiaenen, Katrijn Klingels.

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
