## [Decision Letter · Decision Letter 0]

16 Feb 2023

PONE-D-22-31852How do therapists teach motor skills to children with Developmental Coordination Disorder? An interview studyPLOS ONE

Dear Dr. van der Veer,

Thank you for submitting your manuscript to PLOS ONE. After careful consideration, we feel that it has merit but does not fully meet PLOS ONE’s publication criteria as it currently stands. Therefore, we invite you to submit a revised version of the manuscript that addresses the points raised during the review process. Along with the reviewers, I find value in the work reported here and consider your work to be relevant to research and practice. I believe that the points raised by the reviewers would help in ensuring clarity and better dissemination of your work. Thus, we look forward to seeing your revision. 

We look forward to receiving your revised manuscript.

Kind regards,

Catherine M. Capio

Academic Editor

PLOS ONE

Journal Requirements:

2. Your abstract cannot contain citations. Please only include citations in the body text of the manuscript, and ensure that they remain in ascending numerical order on first mention.

Reviewers' comments:

Reviewer's Responses to Questions

**Comments to the Author**

1. Is the manuscript technically sound, and do the data support the conclusions?

Reviewer #1: Yes

Reviewer #2: Yes

Reviewer #3: Yes

2. Has the statistical analysis been performed appropriately and rigorously? 

Reviewer #1: N/A

Reviewer #2: Yes

Reviewer #3: Yes

3. Have the authors made all data underlying the findings in their manuscript fully available?

Reviewer #1: Yes

Reviewer #2: Yes

Reviewer #3: Yes

4. Is the manuscript presented in an intelligible fashion and written in standard English?

Reviewer #1: Yes

Reviewer #2: Yes

Reviewer #3: Yes

5. Review Comments to the Author

Reviewer #1: This work demonstrates an important contribution to knowledge. I have suggested some revisions below for your consideration to ensure greater clarity for the reader.

It would be good to outline in the abstract that it was only Physiotherapists included in the sample.

Provide a justification for why Physiotherapists were the target audience as you mention Occupational Therapists as well in your introduction.

Consider changing the title to make it clear only Physiotherapists were included in the sample.

Line 51 and 52 - I suggest re-wording as it is unclear why the sentence about task and environmental adaptation starts with a ‘but’ as if in contrast to something. It would also be good to provide an example of this adaption.

From line 53 when describing how mlm can be adapted can you provide an example

Line 64 - a little more detail on what ‘intuitive-driven mls’ is and how this contrasts to other types, do the other types have a name?

From Line 78 a little more detail on COOP and NTT needed and how/if these can be classified as MLS. It is a little confusing because you go on to say “a limited amount is known about the effectiveness of MLSs in children with DCD.” but there is quite a lot of evidence for COOP and NTT. This needs clarifying.

This sentence is confusing “In summary, MLSs are fundamental for teaching motor skills to children with DCD. However, a limited amount is known about their effectiveness” - how can they be fundamental if there is limited knowledge about their effectiveness?

Make clear if data from the pilot interviews was discarded or used in the analysis

You mention “(probable) DCD” - it would be good to provide more information on how therapists were asked to identify the children they reflected on or worked with that informed this study.

147 - A little more clarity on how many interviews their were in total and where they were all held.

Greater clarity needed on how many people had interviews, and how many participated in focused groups and what cross over was - a lot of the information in ‘results’ -”Process of recruitment and data collection” should be moved up for clarity.

Make the rational clearer for how/why you decided what information should be gathered in interviews and which should be gathered by focus groups and why both were needed.

Explain what made you think that you reached ‘data saturation’ was was the evidence?

Explain what errorless learning, analogy learning and motor imagery are.

Rather than using ‘it’ when discussing children it is better to use ‘they’

Line 264 - “Therapists frequently mentioned the following child characteristics as guiding their choices” - can you say how this guided their choice?

I suggest rewording this category “main problem of children with DCD” - as it is overly negative and vague.

Line 363 - “Therapists enlarged” - typing error

Paragraph 412 needs rewording so that the argument being made is clear.

Line 437 “DCD, because of their learning disabilities” - not all children with DCD have additional ‘learning disabilities’ - this needs rewording for clarity.

Could you write the discussion more concisely?

Line 466 did this review include studies on children with DCD?

504 - “leaving them with no opportunity to videotape treatment sessions” change this wording as therapists did submit videos

Line 513 “ the identified child 513 characteristics appeared generalizable to other types of children as well.” - this needs further explanation. This links to an earlier point about clarifying how therapists were asked to identify the children they were reflecting on.

Reviewer #2: Dear Authors

As far as I can judge you have written a very ambitious manuscript and I suggest a minor revision.

My concerns are as follow.

# Three small editorial notes, p.7 lines 135 and 137 I would suggest teach/ing instead of train/ing. P.24, line 507, I would suggest Further instead of Secondly. P.24, lines 483, 484 and 486 please write the whole name not only the abbreviations NTT and CO-OP.

# The MLS's presented are both extremely cognition centered. Through the manuscript there are connected concepts such as strategies, instructions, teaching, learning, explicit. The participants report that they are lacking knowledge and have need for more education. Some report that they have difficulties teaching how to skip and how to skip rope and still another suggested that, in order to teach a child to ride a bike, let it ride down hill. When it came to automatization and transfer (p.15) participants said that lack of motivation creates frustration. With all respect, I wonder if the participants in this study lack basic knowledge about child development. From my point of view the reported interventions are too much top-down approaches, which in the long (and even short)run are too difficult for children diagnosed DCD. However, this doesn't change my impression of your study. The results are what they are and, still, from my point of view you have done a good work.

However (again), I would suggest you to include a short comment, in the Discussion, on the participants experiences in connection to limits of top-down approaches. I would also suggest you to mention that there are approaches that are mostly bottom-up, which have reported good results e.g., Niklasson et al., 2017 'Catching-up..' This said because I feel pity for the participants since I read between the lines (I might be wrong) that they obviously are unsatisfied with their results both during and after training. Perhaps they would benefit from knowledge about/education in bottom-up approaches. It is always good to have many tools in a tool box. As for now the tools are Instructions, which in a way is equal to in-form i.e., bring in from outside. What might be needed, and what I suggest, is Education, edu-care, i.e., bring out what is inside. That is, start at a basic level i.e, with what the child and its body knows. There would be less frustration and better motivation.

# Among the participants there were only one man. I would appreciate to read what you think about that. Are there differences in how women approach the issues reported in your study compared to men? If you find it appropriate, add a short comment in the Discussion.

Reviewer #3: General Comments:

This is valuable information for those working in the areas of DCD and motor learning. I applaud you for taking on the challenging work of doing qualitative research. As you will see below, the major comments related to the article are primarily in the Methods and Results sections.

Introduction: Very nice overview.

Lines 49-51: You have suggested types of practice. Is there a reason why parts practice and whole practice are not included here? It seems that there were comments from therapists about that as it is a commonly used MLS.

Line 65 and 88: Since questions you asked of therapists included the environment, it may be helpful to include the environment in these sentences, such as “….the characteristics of the child, the tasks practiced and environmental characteristics.”

Consider a broader purpose statement, such as: “The purpose of this qualitative interview study is to identify and describe factors that influence a therapists use and choice of MLSs when intervening with children with DCD.” This allows you to consider more fully the therapists’ knowledge, preferences and beliefs, but would require some modifications, particularly in the discussion section.

If a final conclusion of this study is that there is a lack of knowledge about MLSs, could you contribute to acquisition or exposure to MLSs by including a table of MLSs in the introduction, perhaps with descriptions and article references (it could also be in a supplemental appendix). It may help the reader with a better understanding of interview results and discussion.

The table could look something like this:

Table *** Motor Learning Strategies Commonly Described in Literature

Concepts, Instruction and Feedback Descriptions

• Internal and External Feedback

• Implicit and Explicit motor learning

• Errorless and analogy learning

• etc

Organization of Practice

• Variable and Constant Practice

• Random and Blocked Practice

• Whole and Parts Practice

• etc

Materials and Methods:

Most of these comments relate to clarity in the methods so that another researcher would be able to replicate your study.

Line 96: Begin with a clear statement about the type of study this is.

Line 108: Consider changing “record” to videotape as “record” could also be confused with audiorecording

Line 137: Explain “topic of interest.” Would that be DCD and motor learning strategies?

Line 154: Could you provide a statement about why the questions were different in focus group 1 and 2?

Lines 170-176 and Lines 211-217: There is overlap in some content related to number of interviews, timing and saturation. Consider which section is most appropriate and omit some content in the other section.

Results:

Line 220-221: For readers in different health care systems, could you provide a brief explanation (perhaps even in parentheses) of primary and secondary health care.

Table 2: f/m is Sex, not Gender (socially constructed roles)

Findings of the interview analysis:

The value of a qualitative interview study is the quotations from those interviewed. The Themes under the Interview Analysis section is not well organized. It is not clear what each of the individual characteristics clearly reflect. It could be presented in a much more organized manor.

To have completed the coding and decided on 6 themes and 21 characteristics, it follows that there should be at least 2 quotations/text fragments per characteristic that you could use to clarify the importance or meaning of those characteristics.

If that is true and you can identify at least 2 clarifying quotes or text fragments for each characteristic, then you could improve your organization and clarity of this section of the manuscript and perhaps provide even more clarity with a table. The text explanation of each characteristic could include “the best” quote and a second one could be included in a reconstructed table.

Line 243+

Begin by succinctly introducing the Theme, then continue with subheadings for each characteristic with an explanation of that characteristic and a quote that relates to that characteristic.

For example:

Theme

- Introduction of theme

Characteristic (it would be best to have a sub-heading for each one)

- Provide an explanation of characteristic

- Provide a quote/text fragment that relates to the characteristic

Here is a suggestion for a reconstruction of Appendix A-Fig 2 that may include the following

Table *** Themes, Characteristics and Supportive quotations

Themes Characteristics Supportive quotation

Tailor-made treatment

1

2

3

4

5

Therapists’ teaching style

1

2

3

Etc Etc Etc

Discussion:

Line 381: The opening statement should reiterate the purpose statement if you change the purpose statement based on the comment above.

The Discussion could be organized for more clarity, focusing on each of the major influences on clinical decision making. Perhaps consider reorganizing with sub-headings such as:

• Child characteristics

• Tasks practiced

• Environmental influence

• Therapist characteristics

• Strengths and limitations

• Conclusions

Minor comments:

Abstract Line 27: (1) should be “treated” to stay consistent with past tense of 2-6

It is not clear why and when you use the word therapists vs PPTs. Please be consistent.

6. PLOS authors have the option to publish the peer review history of their article (what does this mean?). If published, this will include your full peer review and any attached files.

Reviewer #1: No

Reviewer #2: **Yes: **Mats Niklasson

Reviewer #3: No

---

## [Author Response · Author response to Decision Letter 0]

4 Oct 2023

Author’s response to decision letter for PONE-D-22-31852

How do therapists teach motor skills to children with Developmental Coordination Disorder? An interview study

Diepenbeek, 31-03-2023

Dear editor in chief, dear Prof. Dr. C. Capio, 

Please find uploaded our revision of the manuscript entitled: “How do therapists teach motor skills to children with Developmental Coordination Disorder? An interview study” 

We revised our manuscript as requested in your email of February 16, 2023.

We would like to thank the editor and the reviewers for their constructive and detailed feedback, and for giving the opportunity to revise our manuscript. Below we provide a point-to-point reply to each of the comments (in italic blue coloured). In the manuscript, we highlighted the changes by colouring the text blue, and the deleted text that was not replaced with track-changes. 

We hope that with this revision and reply all concerns are satisfactorily addressed and the manuscript can be accepted for publication. Of course, we will be happy to answer any additional questions from the editorial office or reviewers. 

Yours sincerely,

Ingrid van der Veer (first author, on behalf of all co-authors)

General comments:

2. Your abstract cannot contain citations. Please only include citations in the body text of the manuscript, and ensure that they remain in ascending numerical order on first mention.

Reply: the manuscript meets the style requirements.

Reviewer comments:

Reviewer #1

This work demonstrates an important contribution to knowledge. I have suggested some revisions below for your consideration to ensure greater clarity for the reader.

Thank you for your detailed feedback.

It would be good to outline in the abstract that it was only Physiotherapists included in the sample.

Reply: in the methods of the abstract was mentioned that we included paediatric physical therapists (PTTs). In the results and conclusion of the abstract we used ‘therapists’. For more clarity, we changed therapists into PPTs in the results and conclusion.

Reviewer #3 also suggested to be more consistent in the terms PPTs and therapists. In order to make it clearer for the reader, we decided to change therapists into PPTs in the entire manuscript. 

Provide a justification for why Physiotherapists were the target audience as you mention Occupational Therapists as well in your introduction.

Reply: we focused on PPTs because in the Netherlands and Flanders children with DCD are frequently treated by PPTs. Furthermore, it was a logical choice to include either PPTs or occupational therapists from a methodological point of view. Because we changed therapists into PPTs, a justification in the text was not needed anymore.

Consider changing the title to make it clear only Physiotherapists were included in the sample.

Reply: we changed title.

This now reads: “How do paediatric physical therapists teach motor skills to children with Developmental Coordination Disorder? An interview study”

Line 51 and 52 - I suggest re-wording as it is unclear why the sentence about task and environmental adaptation starts with a ‘but’ as if in contrast to something. It would also be good to provide an example of this adaption.

Reply: we rephrased the sentence and added an example.

This now reads: “Therapists can manipulate task and environment to enhance motor learning as well, for instance, by decreasing distance to the target to improve throwing beanbags into a basket”

From line 53 when describing how mls can be adapted can you provide an example

Reply: we added an example

This now reads: “It advocates that MLSs should be adapted to the unique combination of characteristics to address the specific needs and capacities of the individual child, for example, therapists simplify cognitive load by adopting less stringent task rules or using observational instructions [7].”

Line 64 - a little more detail on what ‘intuitive-driven mls’ is and how this contrasts to other types, do the other types have a name?

Reply: we meant that they used MLSs intuitively instead of based on theory or scientific evidence. We rephrased the sentence.

This now reads: “However, in another study, investigating PTs’ perspectives on the construct of motor learning, PTs stated that they had difficulty understanding the theoretical construct of motor learning, and that their knowledge was limited, steering them towards an intuitive use of MLSs [10].”

From Line 78 a little more detail on COOP and NTT needed and how/if these can be classified as MLS. It is a little confusing because you go on to say “a limited amount is known about the effectiveness of MLSs in children with DCD.” but there is quite a lot of evidence for COOP and NTT. This needs clarifying.

Reply: NTT and CO-OP are intervention methods in which therapists use MLSs to enhance children’s motor learning (Missiuna et al. 2001 and Schoemaker et al. 2003). We agree that evidence shows that NTT and CO-OP are effective. However, in the methods of studies investigating effectiveness of CO-OP and NTT it is not described and clarified what MLSs are used. So, these studies provide insight into the effect of the intervention on motor performances and goals, but little into effectiveness of individual MLSs. 

Studies investigating effectiveness of individual MLSs in children with DCD are limited and results are conflicting or no differences between groups were found (Bonney et al. 2017a, Bonney et al. 2017b, Subara-Zukic et al. 2022). Thus, it remains unclear which MLSs should be used best in children with DCD.

We added details on NTT and CO-OP for clarification.

This now reads: “These child-centred interventions are based on theories of motor learning, and MLSs derived from motor learning research are used to manipulate the interaction between child, task and environment to improve motor skills [17,18]. However, a limited amount is known about the effectiveness of individual MLSs used in children with DCD.”

This sentence is confusing “In summary, MLSs are fundamental for teaching motor skills to children with DCD. However, a limited amount is known about their effectiveness” - how can they be fundamental if there is limited knowledge about their effectiveness?

Reply: we agree that the chosen phrasing is confusing. MLSs are considered fundamental in the NTT and CO-OP interventions (Missiuna et al. 2001 and Schoemaker et al. 2003), and this was also mentioned by PTs in an interview study exploring PTs’ perspectives on motor learning (Atun-Einy et al. 2019). But from the research perspective, there is little known about the effectiveness of MLSs used in children, and in children with DCD more specifically (Bonney et al. 2017a, Bonney et al. 2017b, Subara-Zukic et al. 2022, van Abswoude et al. 2021, Simpson et al. 2020). 

We rephrased the sentence for more clarity.

This now reads: “In summary, in activity- or participation-oriented interventions the use of MLSs is considered fundamental for teaching children with DCD motor tasks. However, …”

Make clear if data from the pilot interviews was discarded or used in the analysis

Reply: the data was discarded, we added this to the method section.

This now reads: “After three pilot interviews with members of the target population, the interview guide was finalised (S2 File). The data from the pilot interviews were discarded.”

You mention “(probable) DCD” - it would be good to provide more information on how therapists were asked to identify the children they reflected on or worked with that informed this study.

Reply: this is described in more detail in the other article regarding the results of the observations which is accepted for publication by the journal “Physical and Occupational Therapy in Pediatrics”. 

We now added more detail in here.

This now reads: “… with (probable) DCD. Preferably, the child was diagnosed with DCD. However, because the mean age of receiving a diagnosis of DCD in the Netherlands is 7.02 years (SD 1.79), and the process of diagnosis takes an average of 2.79 years (SD 2.13 years) [24], PPTs were able to video-tape a treatment session of a child with probable DCD if the child met all four DSM-5 criteria for DCD.”

147 - A little more clarity on how many interviews there were in total and where they were all held.

Reply: five of 12 interviews were conducted online due to Covid-19. The remaining 7 interviews were performed physically at the PPTs’ own workplace. We rephrased the sentence for more clarity.

This now reads: “The interviews were held in the PPTs’ own workplace so that they could support their answers with demonstrations. However, due to Covid-19 restrictions, five of 12 individual interviews were conducted online with Skype or Google Meet [25].”

 

Greater clarity needed on how many people had interviews, and how many participated in focused groups and what cross over was - a lot of the information in ‘results’ -”Process of recruitment and data collection” should be moved up for clarity.

Reply: based on previous point of feedback, we added the total amount of individual interviews to the text. We also added that we performed two focus groups to the section ‘Focus-groups interviews’. We described when they were planned, and that none of the PTTs participated in the individual interviews. 

This now reads: “Two focus-group interviews were planned: the first one after 10 individual interviews, and the second after the final individual interview. None of the participating PPTs had participated in the individual interviews.”

In the section ‘Data collection’ the flow of the interviews was already described. However, based on your feedback in a subsequent point of feedback regarding saturation, we rephrased this sentence for more clarity.

This now reads: “Focus group 2 was organized when data saturation in the individual interviews seemed reached. This was the case when two consecutive individual interviews identified no new themes, and provided no new meaningful information to better understand the identified themes [22,28].”

Make the rational clearer for how/why you decided what information should be gathered in interviews and which should be gathered by focus groups and why both were needed.

Reply: in the section ‘Individual interviews’ we added our reason for using individual interviews in the first sentence of the paragraph.

This now reads: “Individual interviews were conducted to gain insight into the individual reasons of PPTs about their choices in MLSs used to teach motor tasks to children with DCD [21,23].”

In the section ‘Focus-groups interviews’ we had already mentioned that the aim of the focus groups was three-fold (1. deepen topics, 2. elaborate on different point of views, and 3. determine whether obtained insights were shared by a larger group). For clarity, we added that we included focus groups to enrich the data and rephrased the sentence about the reasons to make clearer what they added.

This now reads: “Focus groups were conducted in addition to the individual interviews to enhance data richness [21,26]. The focus-group interviews: (1) deepened topics mentioned in the individual interviews; (2) clarified and elaborated on different points of view about the use of MLSs; and/or (3) determined that insights obtained from the individual interviews were shared by a larger group of therapists [21,26].” 

Explain what made you think that you reached ‘data saturation’ what was the evidence?

Reply: saturation was reached because Individual interviews 11 and 12, and Focus group 2, identified no new themes anymore, and the data provided no new meaningful information to better understand the identified themes. We added this to the text in section “Methods – Data collection”.

This now reads: “Focus group 2 was organized when data saturation in the individual interviews seemed reached. This was the case when two consecutive individual interviews identified no new themes, and provided no new meaningful information to better understand the identified themes [22,28].”

Explain what errorless learning, analogy learning and motor imagery are.

Reply: Reviewer #3 suggested to provide descriptions (with references) of MLSs to support the reader. We provided these descriptions in Supplemental File 1 and referred to it in the introduction. In the results, where we mentioned errorless learning, analogy learning and motor imagery, we referred to this S1 File as well.

In the introduction this now reads: “See S1 File for (more) descriptions of MLSs commonly described in literature.”

In the results this now reads: “Concepts, such as implicit and explicit motor learning, and specific learning strategies, like errorless learning and analogy learning (S1 File), were not explicitly known by the majority of the PPTs.”

Rather than using ‘it’ when discussing children it is better to use ‘they’

Reply: we carefully re-read the text and made some changes where appropriate. However, mostly it concerned quotes, and we decided not to change wording of quotes because then it would not represent the words of the PPTs anymore.

Line 264 - “Therapists frequently mentioned the following child characteristics as guiding their choices” - can you say how this guided their choice?

Reply: in general, the data showed large variation in how specific child characteristics influenced PPTs choices, and this aspect needs further research. Therefore, we decided to provide here a short overview of the mentioned characteristics and only elaborated on specific child characteristics when it supported other themes. In our opinion, discussing the variation for each characteristic would have added little value to this paper. We rephrased the beginning of the paragraph, and added that variation in preferred MLSs was large.

This now reads: “Several child characteristics were mentioned when PPTs elaborated on their choice for MLSs. However, variation in preferred MLSs for specific child characteristics was large. Some of the child characteristics will be discussed in more detail in the next themes. Following child characteristics were mentioned frequently: …” 

 

I suggest rewording this category “main problem of children with DCD” - as it is overly negative and vague.

Reply: we agree that “main problem of…” sounds overly negative. We changed the category into “specific learning disabilities in children with DCD” in the text and in Table 3 (which replaced Figure 2).

This now reads: “This theme consisted of three categories: (1) specific learning disabilities of children with DCD; (2) strategies to improve time spent on task; and (3) strategies to enhance transfer.

Most PPTs referred to the problems with automatization and transfer of skills as the specific learning problems in children with DCD.”

Line 393 - “Therapists enlarged” - typing error

Reply: this should have been ‘elaborated’.

This now reads: “PPTs elaborated on how specific child characteristics and/or the task practised guided their clinical decision-making.”

Paragraph 412 needs rewording so that the argument being made is clear.

Reply: for clarity, we made three changes to this paragraph to make the argument clear:

1. We added that PPTs had different assumptions on whether executive functions could be trained or not. 

2. We changed the order in the second sentence because it seemed more logical.

3. We added to the final sentence of the paragraph that executive functions can be trained, and that PPTs can adopt an indirect teaching style to advance the development of executive functions.

This paragraph now reads: “One child characteristic that PPTs frequently mentioned guiding their choice of an indirect or direct teaching style was the presence of deficits in executive functions. As a result of deficits in inhibitory control, working memory, and attention, children with DCD have problems in planning and organizing activities of daily life [7,12]. Based on their assumption whether executive functions could be trained or not, some PPTs preferred an indirect teaching style, while others preferred a direct style. Research shows that executive functions in children can be improved by training [34,35]. Advancing critical and creative thinking, and problem-solving in movement situations encourages the development of executive functions [36]. Using questions that require children to think about movement solutions and then debriefing them about their actions is a frequently used strategy. Another is to place children in movement situations that challenge them to think about movement solutions [36]. Both strategies were used by the PPTs with indirect teaching styles. Because executive functions are important in many daily life activities (e.g. in learning at school, and in social interactions), and can be trained, PPTs can adopt an indirect teaching style to enhance the development of these executive functions when teaching motor tasks [36,37].”

Line 437 “DCD, because of their learning disabilities” - not all children with DCD have additional ‘learning disabilities’ - this needs rewording for clarity.

Reply: we rephrased the sentence.

This now reads: “PPTs stressed that attention to motivation is specifically needed in children with DCD, because most children experience problems in motor learning and have lower levels of perceived competence.”

Could you write the discussion more concisely?

Reply: it is not clear what the reviewer exactly would like to see changed. We re-read the discussion and rephrased various sentences to make the text more concise. Furthermore, we added subheadings to structure the discussion, and re-organized paragraphs based on a suggestion of Reviewer #3. Specifically, the sections regarding the main findings contain multiple changes. 

This now reads:

“Factors guiding PPTs’ process of clinical decision making

PPT characteristics

Most PPTs experienced difficulties putting their thoughts into words about which characteristics led their clinical decision-making. They stressed that the interaction of child, task, and environment guided their choices, as suggested by the 'hybrid model of DCD’ [7]. However, the results of this study showed that PPTs choose MLSs intuitively, and that their clinical decision-making process was influenced by their own characteristics like knowledge, preferences, and beliefs as well. These findings are in line with a previous interview study that explored PTs’ perspectives on the construct of motor learning and their experiences of its implementation in clinical practice [10]. PTs stated that their use of MLSs was guided by intuition, and that their limited knowledge was an important barrier to implementation [10]. The importance of knowledge has also been demonstrated in several think-aloud studies investigating PTs’ clinical decision-making processes in rehabilitation, showing that knowledge from prior clinical experience, education, scientific research, and mentors or colleagues influenced their clinical decision-making [8,32,33]. For optimal clinical decision making, PPTs require knowledge about: the use of MLSs to teach motor tasks (including adapting MLSs to child and task); the learning disabilities and associated problems of children with DCD; and basic knowledge about child development. The results of this study indicates the importance of the level of education on these topics. Specifically, PPTs’ knowledge about implicit and explicit motor learning approaches appeared limited. A need for more education has also been stressed by previous research [8,10].

Child characteristics

PPTs elaborated on how specific characteristics of a child guided their use of MLSs. Various child characteristics were identified. However, because of large variation in suggested characteristics and preferred MLSs, more research is required to gain insights into how the identified characteristics can guide PPTs’ decisions. One child characteristic that PPTs frequently mentioned guiding their choice of an indirect or direct teaching style was the presence of deficits in executive functions. As a result of deficits in inhibitory control, working memory, and attention, children with DCD have problems in planning and organizing activities of daily life [7,12]. Based on their assumption whether executive functions could be trained or not, some PPTs preferred an indirect teaching style, while others preferred a direct style. Research shows that executive functions in children can be improved by training [34,35]. Advancing critical and creative thinking, and problem-solving in movement situations encourages the development of executive functions [36]. Using questions that require children to think about movement solutions and then debriefing them about their actions is a frequently used strategy. Another is to place children in movement situations that challenge them to think about movement solutions [36]. Both strategies were used by the PPTs with indirect teaching styles. Because executive functions are important in many daily life activities (e.g. in learning at school, and in social interactions), and can be trained, PPTs can adopt an indirect teaching style to enhance the development of these executive functions when teaching motor tasks [36,37]. PPTs choices based on the characteristics learning stage, the presence of learning disabilities, level of motivation, and level of perceived competence will be discussed in the section ‘Key elements in motor learning’. 

Task complexity

Task complexity was identified as the most important task characteristic guiding PPTs’ choice for MLSs. It appeared a complex construct described by four variables: (1) task constraints; (2) environmental demands; (3) child characteristics; and (4) therapist characteristics. The ‘challenge point framework’ conceptualizes complexity as a result of the combination of child, task and environment to which the PPTs should adapt their MLSs, which is in line with the results of our study [38]. The framework distinguishes two types of difficulties: the nominal task difficulty is defined by the task constraints, and is considered to reflect a constant amount of difficulty; the functional task difficulty is determined by the experiences of the individual (e.g. novices experience tasks as more difficult than individuals who have already performed those tasks) and environment (e.g. throwing outside in windy circumstances is more challenging than throwing indoors) [38]. This study also demonstrated that PPTs can experience teaching specific motor tasks as more or less difficult based on their knowledge and experiences.

PPTs’ opinions on which MLSs to use in complex tasks (e.g. riding a bike, tying shoe laces, and writing) differed: some used specific instructions focusing on the planning of these motor tasks, while others chose to provoke the correct movements by manipulating task and context. According to the international DCD recommendations, evidence-based methods like CO-OP and NTT can be used to teach motor tasks to children with DCD: CO-OP focuses mostly on motor planning, while NTT focuses on manipulating task and context [16–18]. Some therapists explained that they chose specific MLSs because they were trained in CO-OP or NTT. However, other reasons for choosing to focus on motor planning in complex tasks were given as well, demonstrating that PPTs own characteristics influenced their choices: (1) PPTs did not know how to manipulate complex tasks and its context; (2) it suited their own preference in learning; and (3) they believed that children needed to learn motor planning to advance learning in daily life. 

Key elements for motor learning

In the treatment of children with DCD, PPTs gave specific attention for children’s motivation (Key element 1), the optimal practice level (Key element 2), adequacy of time on tasks (Key element 3), and transfer (Key element 4). The key elements motivation and the optimal practice level were related: if the practice level was too difficult or too easy, motivation decreased and learning was hampered. PPTs considered the child’s learning stage when estimating the optimal practice level but their opinions on the use of explicit instructions and feedback in the early learning stage differed: some argued that children needed explicit information to learn tasks that they had not yet mastered, while others said that they reduced the amount of explicit information given because children with DCD experience difficulties with processing large amounts of information. Studies investigating effectiveness of explicit and implicit instructions and feedback used to teach functional motor skills to inexperienced children with DCD found conflicting evidence [39–41]. Systematic reviews investigating the effectiveness of these types of instructions and feedback in children with and without motor disabilities also found conflicting results [42,43]. Thus, both explicit and implicit instructions might be used. 

PPTs stressed that attention to motivation is specifically needed in children with DCD, because most children experience problems in learning motor tasks and have lower levels of perceived competence. Research confirms that both characteristics are prominent in children with DCD [12,13]. The role of motivation in enhancing motor learning is conceptualized in the ‘Optimizing Performance through Intrinsic Motivation and Attention for Learning’ (OPTIMAL) theory [44]. According to this, motivation will be improved by giving autonomy to children, and by enhancing their self-confidence [44]. The findings of the current study showed a large variation between PPTs in strategies used for improving motivation (Table 3). All of them used positive encouragements and experiences of success to enhance self-confidence, in line with the OPTIMAL theory. Only a few enhanced autonomy by giving children choice. Furthermore, some stressed the importance of enjoyment to increase motivation. A systematic review investigating effectiveness of MLSs related to the OPTIMAL theory that enhanced children’s motivation showed that, in most included studies, motor performance improved more when MLSs that enhanced motivation were used compared to MLSs that did not [43]. However, no such studies were performed in children with DCD. Furthermore, the authors reported that: (1) most investigated MLSs focused on feedback; (2) not all MLSs investigated had good ecological validity; and (3) effectiveness seemed modified by child characteristics like motor abilities, and the task practised. They recommended that future studies should explore how MLSs enhancing motivation could be integrated into children’s motor learning [43]. The suggested MLSs in this study could be informative for researchers investigating effectiveness of MLSs. 

PPTs considered the key elements adequacy of time on tasks and transfer important during treatment, because of the specific learning disabilities of children with DCD, and their consequences on the level of participation. Again, the MLSs suggested to improve time spent on task and transfer varied widely between PPTs. Most PPTs highlighted the importance of instructing parents and teachers to practise in the child’s daily context, which is in line with the international DCD recommendations [16]. These recommendations also stress to practice meaningful activities fitting children’s needs, and to consider practising in small groups [16]. Both were mentioned by PPTs as strategies to enhance transfer. Furthermore, PPTs frequently mentioned using variation in practice to enhance transfer, specifically as learning progressed. Studies in children with DCD showed no differences in the effectiveness of variable versus constant practice on immediate transfer tests after Wii Fit training [19,20]. However, a systematic review including a meta-analysis concluded that effectiveness of variable practice in predominantly healthy young adults seemed promising, but that the included studies were at a high risk of bias, had small sample sizes and were difficult to compare due to large amounts of heterogeneity [5]. The authors also mentioned that variable practice can increase enjoyment, and that it suits real-world contexts better [5]. Both arguments were also raised by therapists in our study.“

Line 466 did this review include studies on children with DCD?

Reply: no, this review included mainly healthy young adults. We added this to the text.

This now reads: “A systematic review including a meta-analysis concluded that effectiveness of variable practice in predominantly healthy young adults seemed promising, …”

504 - “leaving them with no opportunity to videotape treatment sessions” change this wording as therapists did submit videos

Reply: we rephrased this sentence.

This now reads: ”Firstly, recruitment was challenging for several reasons: (1) PPTs found participation too time-consuming; (2) for a period of time, they had no opportunity to videotape treatment sessions because they were not allowed to treat children due to Covid-19; and (3) …”

Line 513 “ the identified child 513 characteristics appeared generalizable to other types of children as well.” - this needs further explanation. This links to an earlier point about clarifying how therapists were asked to identify the children they were reflecting on.

Reply: PPTs might have shared experiences in teaching motor tasks to other types of children then DCD. We added the sentence about the identified child characteristics to demonstrate that this limitation had low impact according to our opinion, because most suggested child characteristics mentioned by the PPTs are generic (e.g. age, level of motivation, learning stage). However, after re-reading the text, this is not that clear. We decided to remove this sentence, and leave the judgement about the impact to the reader.

 

Reviewer #2

Dear Authors

As far as I can judge you have written a very ambitious manuscript and I suggest a minor revision.

Thank you for your feedback

My concerns are as follow.

# Three small editorial notes, p.7 lines 135 and 137 I would suggest teach/ing instead of train/ing. P.24, line 507, I would suggest Further instead of Secondly. P.24, lines 483, 484 and 486 please write the whole name not only the abbreviations NTT and CO-OP.

Reply: thank you for these suggestions. We adopted the first and second ones. Because NTT and CO-OP were written as whole name in the introduction, we decided to use the abbreviations in the discussion.

In the section ‘Individual interviews’, this now reads: “Six students each received 35 hours of education to make them familiar with the interview guide and procedure, and to teach them in interview skills such as using prompts and probes. Education included: …”

In the section ‘Data analyses’, this now reads: “… were analysed by the first author (IvdV), together with a master’s student. Three students each received 25 hours of education to acquire analysing skills and to standardize the procedure. Education included: …”

In the section ‘Discussion – Strengths and limitations’, this now reads: “Further, despite intensive recruitment efforts, a Flemish focus group could not be included.”

# The MLS's presented are both extremely cognition centered. Through the manuscript there are connected concepts such as strategies, instructions, teaching, learning, explicit. The participants report that they are lacking knowledge and have need for more education. Some report that they have difficulties teaching how to skip and how to skip rope and still another suggested that, in order to teach a child to ride a bike, let it ride down hill. When it came to automatization and transfer (p.15) participants said that lack of motivation creates frustration. With all respect, I wonder if the participants in this study lack basic knowledge about child development. From my point of view the reported interventions are too much top-down approaches, which in the long (and even short)run are too difficult for children diagnosed DCD. However, this doesn't change my impression of your study. The results are what they are and, still, from my point of view you have done a good work.

However (again), I would suggest you to include a short comment, in the Discussion, on the participants experiences in connection to limits of top-down approaches. I would also suggest you to mention that there are approaches that are mostly bottom-up, which have reported good results e.g., Niklasson et al., 2017 'Catching-up..' This said because I feel pity for the participants since I read between the lines (I might be wrong) that they obviously are unsatisfied with their results both during and after training. Perhaps they would benefit from knowledge about/education in bottom-up approaches. It is always good to have many tools in a tool box. As for now the tools are Instructions, which in a way is equal to in-form i.e., bring in from outside. What might be needed, and what I suggest, is Education, edu-care, i.e., bring out what is inside. That is, start at a basic level i.e, with what the child and its body knows. There would be less frustration and better motivation.

Reply: systematic reviews investigating various types of interventions used in children with DCD showed that activity- and participation-oriented interventions have stronger evidence then process-oriented interventions (Smits-Engelsman et al. 2018, Preston et al. 2019). Also, the international DCD recommendations advise to use these activity- and participation-oriented interventions (Blank et al. 2019). Therefore, we focused on these in this study. 

In the interviews we asked how PPTs used activity-oriented interventions, and whether they were familiar with motor learning terminology used in literature (which appeared limited). We did not deepen their experiences in connection to the limits of these interventions, and their experiences in process-oriented interventions, because that was beyond the scope of our research question. However, we acknowledge that more research in process-oriented interventions is warranted.

Although, most PPTs expressed the need for more knowledge. We did not have the impression that PTTs were unsatisfied with their results. But we agree, that more knowledge (also about child development) will optimize their use of MLSs. Therefore, we expanded our recommendation on education by adding that education should also include knowledge about the learning disabilities of children with DCD, and basic knowledge on child development.

In the discussion, section ‘PPT characteristics’, this now reads: “For optimal clinical decision making, PPTs require knowledge about: the use of MLSs to teach motor tasks (including adapting MLSs to child and task); the learning disabilities and associated problems of children with DCD; and basic knowledge about child development. The results of this study indicate the importance of the level of education on these topics.”

In the discussion, section ‘recommendation for future research’, this now reads: “This study indicates the importance of the level of education on: using MLSs to teach children motor tasks (including adapting MLSs to child and task), the learning disabilities and associated problems in children with DCD, and child development.”

# Among the participants there were only one man. I would appreciate to read what you think about that. Are there differences in how women approach the issues reported in your study compared to men? If you find it appropriate, add a short comment in the Discussion.

Reply: indeed there was only one male PPT included. According to statistics of the Dutch association for physical therapists (see pdf-document in the link, page 2: Statistics of the Dutch association for PTs), only 6% of the Paediatric PTs registered in the ‘register for quality’ is male. For the Flemish population we have no insights into the ratio male-female PPTs. So, the ratio in our study (3% male) seems representative.

Studies investigating the influence of (P)PTs’ gender on their teaching skills are lacking. However, a study investigating the influence of physical education teachers’ gender on their teaching skills showed that there were no major differences between males and females (Galecki et al. 2011).

We added the presence of one man to the limitations.

This now reads: “Further, despite intensive recruitment efforts, a Flemish focus group could not be included. Also, only one male PPT was included in this study, which seemed a logical consequence of less males working as PPTs. In the Netherlands, only 6% of the PPTs registered in the quality system of the Dutch association of PTs was male [45]. Our research aim was to explore therapists’ use of MLSs, we did not focus on differences between subgroups of PPTs. Despite the recruitment challenges, we had been able to include a heterogeneous sample of PPTs, and reached saturation.”

 

Reviewer #3

General Comments:

This is valuable information for those working in the areas of DCD and motor learning. I applaud you for taking on the challenging work of doing qualitative research. As you will see below, the major comments related to the article are primarily in the Methods and Results sections.

Thank you for your compliments and feedback

Introduction: Very nice overview.

Lines 49-51: You have suggested types of practice. Is there a reason why parts practice and whole practice are not included here? It seems that there were comments from therapists about that as it is a commonly used MLS.

Reply: we mentioned some examples of MLSs for illustration in the introduction, but had no intention to provide a comprehensive overview. You suggested in another point of feedback to provide descriptions with references of the commonly described MLSs in literature. Based on this suggestion we added a supplemental file and referred to it in the introduction. In this file we also described part and whole practice as MLSs for the category organization of practice.

This now reads in the introduction: “See S1 File for (more) descriptions of MLSs commonly described in literature.”

Line 65 and 88: Since questions you asked of therapists included the environment, it may be helpful to include the environment in these sentences, such as “….the characteristics of the child, the tasks practiced and environmental characteristics.”

Reply: we added environment to both sentences.

This now reads: 

- “Thus, PPTs need more insights into how they can choose MLSs based on the characteristics of a child, a task, and an environment.”

- “However, a limited amount is known about their effectiveness, and also about which MLSs to choose, based on characteristics of the individual child, the tasks practised, and the environment.”

Consider a broader purpose statement, such as: “The purpose of this qualitative interview study is to identify and describe factors that influence a therapists use and choice of MLSs when intervening with children with DCD.” This allows you to consider more fully the therapists’ knowledge, preferences and beliefs, but would require some modifications, particularly in the discussion section.

Reply: we agree that your suggested purpose better suits the final results. However, we decided not to rephrase the purpose of this study, because it would not reflect our starting point in this study with the initial research aim. 

If a final conclusion of this study is that there is a lack of knowledge about MLSs, could you contribute to acquisition or exposure to MLSs by including a table of MLSs in the introduction, perhaps with descriptions and article references (it could also be in a supplemental appendix). It may help the reader with a better understanding of interview results and discussion.

The table could look something like this:

Table *** Motor Learning Strategies Commonly Described in Literature

Concepts, Instruction and Feedback Descriptions

• Internal and External Feedback

• Implicit and Explicit motor learning

• Errorless and analogy learning

• etc

Organization of Practice

• Variable and Constant Practice

• Random and Blocked Practice

• Whole and Parts Practice

• etc

Reply: thank you for this suggestion. We added a table in the supporting files with references, and referred to it in the introduction. 

This now reads: “See S1 File for (more) descriptions of MLSs commonly described in literature.”

As a consequence, the current Appendix A and B (interview guides) were changed into S2 File and S3 File.

Materials and Methods:

Most of these comments relate to clarity in the methods so that another researcher would be able to replicate your study.

Line 96: Begin with a clear statement about the type of study this is.

Reply: we added that it was a qualitative study.

This now reads: “In this qualitative study, semi-structured individual and focus-group interviews were conducted to explore how PPTs adapt MLSs to suit children, and how the task being practised influenced their choices.”

Line 108: Consider changing “record” to videotape as “record” could also be confused with audiorecording

Reply: we changed record into videotape.

Line 137: Explain “topic of interest.” Would that be DCD and motor learning strategies?

Reply: indeed, we meant motor learning and DCD. We added this to the sentence.

This now reads: “Education included: reading literature about interviewing and about the topics motor learning and DCD; …”

Line 154: Could you provide a statement about why the questions were different in focus group 1 and 2?

Reply: the topics of the focus groups were based on the analyses of previous interviews. Based on the additional analyses of Focus group 1 and Individual interviews 11 and 12, the focus of the topics for Focus group 2 were more focused compared to the topics of Focus group 1. We added some extra information to the text for clarity.

This now reads: “The findings of the interview analyses prior to the focus-group interviews determined the main topics of these (Table 1). The topics of Focus group 1 emerged from the analyses of the first 10 individual interviews. The topics of Focus group 2 were modified after analysing Focus group 1 and more individual interviews.”

Lines 170-176 and Lines 211-217: There is overlap in some content related to number of interviews, timing and saturation. Consider which section is most appropriate and omit some content in the other section.

Reply: there is indeed overlap. We added the part of the changing focus between the focus groups to the ‘methods – data collection’, and deleted that part in the results. We also rephrased the sentence about the saturation based on feedback of Reviewer #1. 

In the methods this now reads: “An iterative process of data collection and analysis was used to sharpen the focus of the interviews as data collection progressed [27]. As a consequence, the interviews conducted after Focus group 1 focused more on how therapists adapted their MLSs to characteristics of child, task and environment. Focus group 2 was organized when data saturation in the individual interviews seemed reached. This was the case when two consecutive individual interviews identified no new themes, and provided no new meaningful information to better understand the identified themes [22,28].”

In the results this now reads: “After 10 individual interviews, Focus group 1 was organized with eight PPTs to elaborate on six topics gained from findings of the individual interviews (Table 1). Another two individual interviews resulted in no new themes, and provided no new meaningful information for a better understanding of the identified themes. Focus group 2 with six PPTs confirmed saturation (Fig 1).” 

Results:

Line 220-221: For readers in different health care systems, could you provide a brief explanation (perhaps even in parentheses) of primary and secondary health care.

Reply: for clarity, we decided to change it into “private physical therapy practice” and “rehabilitation centre”. We changed it in the results (text and Table 2), and in the methods – recruitment.

In the methods this now reads: “In order to collect a wide range of PPTs’ perspectives, a heterogeneous sample matching the following criteria was required [22]: (1) PPTs with different backgrounds in terms of work settings (e.g. private physical therapy practice, and rehabilitation centre); and (2) …”

In the results, this now reads: “Twenty-three of them worked in a private physical therapy practice, of which six combined this with working in a rehabilitation centre as well. The remaining three PPTs worked in rehabilitation centres (Table 2).”

Table 2: f/m is Sex, not Gender (socially constructed roles)

Reply: we changed Gender into Sex.

Findings of the interview analysis:

The value of a qualitative interview study is the quotations from those interviewed. The Themes under the Interview Analysis section is not well organized. It is not clear what each of the individual characteristics clearly reflect. It could be presented in a much more organized manor.

To have completed the coding and decided on 6 themes and 21 characteristics, it follows that there should be at least 2 quotations/text fragments per characteristic that you could use to clarify the importance or meaning of those characteristics.

If that is true and you can identify at least 2 clarifying quotes or text fragments for each characteristic, then you could improve your organization and clarity of this section of the manuscript and perhaps provide even more clarity with a table. The text explanation of each characteristic could include “the best” quote and a second one could be included in a reconstructed table.

Reply: because of the length of the manuscript we had not added quotes for every category. We added extra quotes to the results section, and provided an extra table with additional quotes which we replaced for Figure 2. Furthermore, we agree that some re-wording and additional sentences will make the categories more visible in the text, and provide greater clarity for the readers. We also highlighted the categories in the text by making them ‘bold’.

We made various changes, we summed them up per theme.

Theme 1: we added two quotes to the text. We decided not to add quotes for the individual child characteristics and task complexity, because they are discussed in more detail (with quotes) in other themes. We did add a sentence that made clear that several characteristics will be discussed in subsequent themes.

This now reads: 

- “All PPTs provided tailor-made treatments to children with DCD. They pointed out that the interaction of child, task and environment most guided their use of MLSs:

“If I look at a child and I see that it is anxious, then that determines how I build my track with exercises. However, if I have a parent that is scared that the child will fall, and reacts negatively every time I let the child jump [of a height], then that will also influence my choices. Furthermore, if a child gets demotivated due to failure, I will change the task. So, I think there is not one [characteristic that is most relevant in making choices].””

- Several PPTs described how, in some cases, it was a search to discover which MLSs worked best. Their main reason for trying different MLSs was that children were experiencing difficulty mastering tasks with one MLS:

“I experienced with this child [the child of the videotape treatment session], that he did not showed improvement. Therefore, I decided to tell him exactly what I expected of him [in the motor performance].”

- “Several child characteristics were mentioned when PPTs elaborated on their choice for MLSs. However, variation in preferred MLSs for specific child characteristics was large. Some of the child characteristics will be discussed in more detail in the next themes. Following child characteristics were mentioned frequently: …”

Theme 2: we only made the categories bold in the text.

Theme 3: we added two quotes, deleted one less relevant sentence (“They also frequently mentioned that children with DCD easily become frustrated during practice, resulting in reduced motivation”), and re-worded the introduction sentence of the second category for greater clarity.

This now reads:

- “Furthermore, PPTs talked about how various child characteristics impacted a child’s motivation according to their opinion. They underpinned the problems in automatization and transfer (Theme 4), and the lower levels of perceived competence of children with DCD as main reasons for these children not being motivated to practice, and getting frustrated when experiencing tasks as being too difficult: 

“If it is really difficult, and it goes wrong every time, I don’t think they [children with DCD] will practice.”

- “The PPTs suggested various strategies to improve children’s motivation (Table 4). For instance, one PPT talked about using themes to improve motivation:

“The boy had no motivation, because he was playing when he had to come to me. I asked what he was doing, and he told me he was making a marble run. So, we drew marble runs when practicing writing readiness skills.””

Theme 4: we rephrased the first category into ‘experiences of success and failure’. Furthermore, we deleted the third category as separate category, because it overlapped with the second category, and added one quote.

This now reads: 

- “Some of them reduced variation during practice to accelerate learning, while others deliberately introduced variation because of the varying contexts found in daily life. In reaction to a PPT that elaborated on how she used various types of ball to stimulate a child’s anticipation abilities in throwing, another PPTs said:

”I practice the basics of the skill [throwing] to make a child familiar with it, and start varying in a later stage.””

 Theme 5: we re-worded Category 1 based on feedback of Reviewer #1 into ‘specific learning disabilities of children with DCD’ (instead of ‘main problem of children with DCD’). Furthermore, we changed Category 2 into ‘strategies to enhance automatization’ (instead of ‘strategies to improve time spent on task’) because the greater time on task had the purpose to enhance automatization. For greater clarity, we split the paragraph in which we mentioned the suggested strategies to enhance automatization and transfer into two separate paragraphs, and added quotes.

This now reads:

- “The PPTs suggested various strategies to enhance automatization. They stressed the importance of instructing parents and teachers to practise in daily life, and underpinned using the same wording in instructions and feedback: 

“They [parents and teachers] should use the same wording as I do, because otherwise they [children with DCD] will never automatize.”

Furthermore, they suggested to practice tasks in similar ways throughout the various treatment sessions, and to decrease instructions and feedback when learning progresses to increase time for repetitions. They felt that with motivated children it was easier to achieve greater time on task.

PPTs also talked about their strategies to enhance transfer. For instance, they varied spatial and temporal constraints during practice (e.g. by continuously changing throwing direction and/or speed to improve the child’s catching abilities) to enhance anticipating to variable contexts in daily life:

”When they [children with DCD] know the movement pattern, than you should start changing to try to simulate other situations [from daily life].”

Theme 6: we added two quotes.

This now reads:

- “The PPTs mentioned following task constraints making tasks more complex: (1) multiple sequential steps; (2) dual tasking; (3) specific timing requirements; (4) bimanual coordination with both hands having different functions; and (5) the requirement to follow rules, for instance, in games. For instance, one PPT said:

“Eating is a bimanual skill, the hands must support each other, while doing opposite tasks” “I think that is what makes eating complex.””

- They also pointed out that environmental demands could increase complexity, for example riding a bike in traffic is much more complex then riding a bike on an empty schoolyard:

“The child could ride a bike inside very well, but he refused to ride outside.” “Riding a bike depends on the person or the environment.”

Line 243+

Begin by succinctly introducing the Theme, then continue with subheadings for each characteristic with an explanation of that characteristic and a quote that relates to that characteristic.

For example:

Theme

- Introduction of theme

Characteristic (it would be best to have a sub-heading for each one)

- Provide an explanation of characteristic

- Provide a quote/text fragment that relates to the characteristic

Here is a suggestion for a reconstruction of Appendix A-Fig 2 that may include the following

Table *** Themes, Characteristics and Supportive quotations

Themes Characteristics Supportive quotation

Tailor-made treatment

1

2

3

4

5

Therapists’ teaching style

1

2

3

Etc Etc Etc

Reply: thank you for your suggestion to replace Figure 2 with a table that includes quotes. We added this table. For now, we decided not to use subheadings per theme, because we expect greater clarity with the changes made based on your previous point of feedback.

Discussion:

Line 381: The opening statement should reiterate the purpose statement if you change the purpose statement based on the comment above.

Reply: we decided not to change the purpose statement. However, we re-organized the discussion as suggested in your subsequent point of feedback. As a consequence, we made minor revisions to the first paragraph of the discussion.

This now reads: ” This qualitative study explored how PPTs adapted MLSs, based on characteristics of a child with DCD, and the task practised. One of the main findings was that PPTs intuitively choose MLSs, and that their clinical decision-making process was not only guided by child and task, but also by their own characteristics (Themes 1 and 6). Another finding was that PPTs used indirect or direct teaching styles, and that they had different justifications for choosing a specific style in children with DCD (Theme 2). Lastly, some general key elements for motor learning in children with DCD emerged when PPTs elaborated on how child characteristics influenced their choices: (1) motivation (Theme 3); (2) optimal level of practice (Theme 4); (3) sufficient time spent on task (Theme 5); and (4) stimulating transfer (Theme 5).”

The Discussion could be organized for more clarity, focusing on each of the major influences on clinical decision making. Perhaps consider reorganizing with sub-headings such as:

• Child characteristics

• Tasks practiced

• Environmental influence

• Therapist characteristics

• Strengths and limitations

• Conclusions

Reply: we added subheadings for clarity and re-organized the section about the main findings. Furthermore, as suggested by Reviewer #1, we rephrased several sentences to make the discussion more concise. 

We added following sub-headings for more clarity.

• Factors guiding PPTs’ process of clinical decision-making

o PPT characteristics: includes the paragraphs about intuition and relevance of knowledge

o Child characteristics: which now includes the paragraph about the direct and indirect teaching styles related to executive functions

o Task characteristics: which now includes the paragraph about the construct of task complexity

See page 12 to 14 of this response letter for how this reads now.

• Key elements for motor learning: we rephrased several sentences to be more concise and to make clearer which key element it concerned. 

See page 14 to 16 of this response letter for how this reads now.

• Strengths and limitations: we added a limitation based on a suggestion by Reviewer #2

• Recommendations for future research: we made a separate section of this, and deleted them from the conclusions

• Conclusions

Minor comments:

Abstract Line 27: (1) should be “treated” to stay consistent with past tense of 2-6

Reply: we changed treat into treated.

This now reads: “(1) PPTs treated children in a tailor-made way;”

It is not clear why and when you use the word therapists vs PPTs. Please be consistent.

Reply: Reviewer #1 also made comments about the use of ‘therapists’ and PPTs’. In order to make it clearer for the reader, we decided to change therapists into PPTs in the entire manuscript.

---

## [Decision Letter · Decision Letter 1]

12 Nov 2023

PONE-D-22-31852R1How do paediatric physical therapists teach motor skills to children with Developmental Coordination Disorder? An interview studyPLOS ONE

Dear Dr. van der Veer,

Thank you for submitting your manuscript to PLOS ONE. After careful consideration, we feel that this revision is a much better version of the manuscript. There are a few remaining editorial points to address, and so we invite you to revise the manuscript further. 

We look forward to receiving your revised manuscript.

Kind regards,

Catherine M. Capio

Academic Editor

PLOS ONE

Journal Requirements:

Additional Editor Comments (if provided):

Thank you for your revision. I agree with the reviewers that this version had improved greatly. There are a few remaining comments, which are mostly editorial in nature that you are now invited to address please.

Reviewers' comments:

Reviewer's Responses to Questions

**Comments to the Author**

1. If the authors have adequately addressed your comments raised in a previous round of review and you feel that this manuscript is now acceptable for publication, you may indicate that here to bypass the “Comments to the Author” section, enter your conflict of interest statement in the “Confidential to Editor” section, and submit your "Accept" recommendation.

Reviewer #2: All comments have been addressed

Reviewer #3: All comments have been addressed

2. Is the manuscript technically sound, and do the data support the conclusions?

Reviewer #2: Yes

Reviewer #3: Yes

3. Has the statistical analysis been performed appropriately and rigorously? 

Reviewer #2: Yes

Reviewer #3: Yes

4. Have the authors made all data underlying the findings in their manuscript fully available?

Reviewer #2: Yes

Reviewer #3: Yes

5. Is the manuscript presented in an intelligible fashion and written in standard English?

Reviewer #2: Yes

Reviewer #3: Yes

6. Review Comments to the Author

Reviewer #2: Dear Authors

I am happy with how you have handled my concerns and I hope your manuscript will be accepted for publication.

Good luck, Mats

Reviewer #3: Thank you for your hard work to revise this manuscript. I believe it is well written and well organized with valuable information. It is a much improved manuscript. I have very few comments below.

Table 3

1)Therapists’ teaching style: PPT characteristics, is there a typo, “Now, IT THINK IT MOSTLY because it suits me.”

2)Motivation: Motivation as a prerequisite for learning is there a typo, “If they fail TO often…..”

Good comments in the Discussion about the OPTIMAL theory as it related to your data. It does not appear that PPT’s reported on one of the third areas of content in the OPTIMAL theory, using an external vs. internal focus of attention. I’m assuming that is why this area was not considered in the Discussion.

Would it be appropriate to add “Enhanced Expectancies” to the Instructions and Feedback section of the SI document as it is one of the pillars of the OPTIMAL theory that is not reflected. Given this theory, I might add to Guided Discovery in S1, by reflecting it as “Autonomy/Guided Discovery.”

7. PLOS authors have the option to publish the peer review history of their article (what does this mean?). If published, this will include your full peer review and any attached files.

Reviewer #2: **Yes: **Mats Niklasson

Reviewer #3: No

---

## [Author Response · Author response to Decision Letter 1]

15 Dec 2023

Author’s response to decision letter for PONE-D-22-31852R1

How do therapists teach motor skills to children with Developmental Coordination Disorder? An interview study

Diepenbeek, 15-12-2023

Dear editor in chief, dear Prof. Dr. C. Capio, 

Please find uploaded our second revision of the manuscript entitled: “How do therapists teach motor skills to children with Developmental Coordination Disorder? An interview study” 

We revised our manuscript as requested in your email of November 13, 2023.

We would like to thank the editor and the reviewers for their feedback, and for giving the opportunity to revise our manuscript. Below we provide a point-to-point reply to each of the comments (in italic blue coloured). In the manuscript, we highlighted the changes by colouring the text blue, and the deleted text that was not replaced with track-changes. 

We hope that with this revision and reply all concerns are satisfactorily addressed and the manuscript can be accepted for publication. Of course, we will be happy to answer any additional questions from the editorial office or reviewers. 

Yours sincerely,

Ingrid van der Veer (first author, on behalf of all co-authors)

General comments:

Journal Requirements:

Reply: We reviewed our reference list and corrected it if needed. Furthermore, we added reference number 22. This paper was not published when we submitted the article, but is now published.

We made two other general changes as well.

• The notation of the affiliation of Hasselt University was incorrect. We changed Faculty of Rehabilitation Sciences and Physiotherapy into Faculty of Rehabilitation Sciences.

• We added the correct data availability statement, as was emailed to you on October 6th 2023.

This now reads:

Following documents are open access available in the UK DataService ReShare repository (10.5255/UKDA-SN-856735): 1. the codebook and memo generated during analyses, 2. the full interview guides of the individual and focus-group interviews, and 3. the informed consent forms. The interview transcripts generated and analysed during this study are not publicly available due to their containing information that could compromise research participants’ privacy/consent. 

 

Reviewer comments:

Reviewer #2

I am happy with how you have handled my concerns and I hope your manuscript will be accepted for publication.

Good luck, Mats

Thank you for the compliments.

Reviewer #3

Thank you for your hard work to revise this manuscript. I believe it is well written and well organized with valuable information. It is a much improved manuscript. I have very few comments below.

Thank you for the compliments and your feedback.

Table 3

1)Therapists’ teaching style: PPT characteristics, is there a typo, “Now, IT THINK IT MOSTLY because it suits me.”

2)Motivation: Motivation as a prerequisite for learning is there a typo, “If they fail TO often…..”

Reply: We corrected the typos

This now reads:

• ... I think it is mostly …

• … fail too often …

Good comments in the Discussion about the OPTIMAL theory as it related to your data. It does not appear that PPT’s reported on one of the third areas of content in the OPTIMAL theory, using an external vs. internal focus of attention. I’m assuming that is why this area was not considered in the Discussion.

Reply: You have interpreted it correctly. The PPTs agreed that attention for motivation was highly relevant in children with DCD, as supported by the OPTIMAL theory. The OPTIMAL theory also suggests to adopt an external focus of attention. However, as mentioned in section r500-509 therapists used both, and literature in children’s motor learning has not yet been able to demonstrate that an external focus of attention benefits children’s motor learning more than an internal focus of attention (e.g. van Cappellen-van Maldegem et al. 2018, Simpson et al. 2020, van Abswoude et al. 2021). 

Would it be appropriate to add “Enhanced Expectancies” to the Instructions and Feedback section of the SI document as it is one of the pillars of the OPTIMAL theory that is not reflected. Given this theory, I might add to Guided Discovery in S1, by reflecting it as “Autonomy/Guided Discovery.”

Thank you for this suggestion, we added “Enhanced Expectancies” to the S1 file. Furthermore, we added “self-controlled” in the sections instruction and feedback, and organization of practice, for the autonomy part of the OPTIMAL theory. As a consequence, an extra reference was added to the reference list, changing the numbers of the references.

This now reads in the Instructions and feedback section:

Instructions and feedback

Self-controlled In self-controlled instructions and feedback the learner determines the timing or modality of the instructions and feedback, which enhances the learner’s autonomy [2].

Enhanced Expectancies Instructions and feedback that enhance the learner’s self-efficacy expectations (or confidence) about the movement performance [2].

In the organization of practice section, this now reads:

Self-controlled In self-controlled practice conditions the learner is provided with choice, which enhances the learner’s autonomy [2].

---

## [Editor Report · Decision Letter 2]

28 Dec 2023

How do paediatric physical therapists teach motor skills to children with Developmental Coordination Disorder? An interview study

PONE-D-22-31852R2

Dear Dr. van der Veer,

We’re pleased to inform you that your manuscript has been judged scientifically suitable for publication and will be formally accepted for publication once it meets all outstanding technical requirements.

Kind regards,

Catherine M. Capio

Academic Editor

PLOS ONE

Additional Editor Comments (optional):

Thank you for your work in this manuscript and the revisions. 
---

## [Editor Report · Acceptance letter]

23 Jan 2024

PONE-D-22-31852R2 

PLOS ONE

Dear Dr. van der Veer, 

I'm pleased to inform you that your manuscript has been deemed suitable for publication in PLOS ONE. Congratulations! Your manuscript is now being handed over to our production team.

Kind regards, 

on behalf of

Dr. Catherine M. Capio 

Academic Editor

PLOS ONE